# Fairness Reprogramming

**Guanhua Zhang**[*]
UC Santa Barbara
guanhua@ucsb.edu

**Yihua Zhang**[*]
Michigan State University
zhan1908@msu.edu

**Yang Zhang**
MIT-IBM Watson AI Lab
yang.zhang2@ibm.com

**Wenqi Fan**
The Hong Kong Polytechnic University
wenqifan@polyu.edu.hk

**Qing Li**
The Hong Kong Polytechnic University
csqli@comp.polyu.edu.hk

**Sijia Liu**
Michigan State University & MIT-IBM Watson AI Lab
liusiji5@msu.edu

**Shiyu Chang**
UC Santa Barbara
chang87@ucsb.edu

## Abstract

Despite a surge of recent advances in promoting machine Learning (ML) fairness, the existing mainstream approaches mostly require training or finetuning the entire weights of the neural network to meet the fairness criteria. However, this is often infeasible in practice for those large-scale trained models due to large computational and storage costs, low data efficiency, and model privacy issues. In this paper, we propose a new generic fairness learning paradigm, called FAIRREPROGRAM, which incorporates the model reprogramming technique. Specifically, FAIRREPROGRAM considers the case where models can not be changed and appends to the input a set of perturbations, called the fairness trigger, which is tuned towards the fairness criteria under a min-max formulation. We further introduce an information-theoretic framework that explains why and under what conditions fairness goals can be achieved using the fairness trigger. We show both theoretically and empirically that the fairness trigger can effectively obscure demographic biases in the output prediction of fixed ML models by providing false demographic information that hinders the model from utilizing the correct demographic information to make the prediction. Extensive experiments on both NLP and CV datasets demonstrate that our method can achieve better fairness improvements than retraining-based methods with far less data dependency under two widely-used fairness criteria. Codes are available at https://github.com/UCSB-NLP-Chang/Fairness-Reprogramming.git.

## 1 Introduction

Fairness in machine learning (ML) has become a critical concern. Due to the biases in data collection, the output prediction is often spuriously correlated with some demographic attributes, which are thus undesirably incorporated into the decision-making process of machine learning models. For example, it is found that some abusive language detection systems tend to classify texts that contain mere mentioning of certain minority groups, *e.g.,* homosexual groups, as abusive content, even though the texts themselves are not abusive at all [1, 2]. Despite the recent advances in fairness promoting learning methods [3–7], the existing mainstreaming approaches mostly require retraining or finetuning the entire model parameters towards an extra fairness objective. However, this is

---

[*]Equal contribution

36th Conference on Neural Information Processing Systems (NeurIPS 2022).

often infeasible in practice, particularly for those well-trained large-scale models, due to the huge computation and storage costs. In addition, for machine learning models that are deployed as a service, model retraining is hindered by limited access to the model parameters.

Recently, model reprogramming has emerged as an alternative technique to model finetuning. In particular, model reprogramming considers the pre-trained model fixed, and instead modifies their input to re-purpose the model towards different objectives. For example, it is shown that a well-crafted input perturbation can re-program an ImageNet classifier to solve the task of counting squares in an image [8, 9]. It is also shown that by learning task-specific embedding prompts concatenated to the inputs, pre-trained language models can achieve better performances than full-parameter tuning in natural language understanding tasks [10–12] Compared with finetuning methods, model reprogramming enjoys lower cost, better scalability, and requires less access to the model parameters. Hence here come our research questions - *Can model reprogramming techniques be applied to fairness objectives? If so, why and how would it work?*

In this paper, we revisit the model reprogramming and propose a novel generic fairness learning paradigm, called FAIRREPROGRAM. In particular, FAIRREPROGRAM perturbs the input by appending to the input a global constant vector/feature, called the *fairness trigger*, which is optimized towards the fairness objective under a min-max framework. FAIRREPROGRAM is a generic framework that works for various tasks and domains. We further introduce an information-theoretic framework that explains why and under what conditions fairness goals can be achieved using a constant fairness trigger. We show theoretically and empirically that the fairness trigger can effectively obscure demographic biases in the output prediction of fixed ML models by providing false demographic information that hinders the model from utilizing the correct demographic information to make predictions.

We perform extensive experiments across various NLP and CV datasets with in-the-wild biases. The results show that FAIRREPROGRAM can consistently achieve better fairness improvement with the retraining-based methods under the two widely-used fairness notions, but with far less trade-off in accuracy. For example, with comparable accuracy, our method can outperform the retraining based baseline with 10.5% and 36.5% lower bias scores over two fairness criteria in the `CelebA` dataset with the hair color prediction task and gender as demographic information. In addition, our method demonstrates great transferability and interpretability. Our theoretical analysis and empirical findings can provide useful insights toward more practical, scalable, and flexible fairness learning paradigms.

## 2   Related Work

**Fairness in ML**   Fairness problems in ML models have received increasing attention from both industry [13] and academia [14–17]. There has been a myriad of fairness definitions in the literature [18, 14, 19, 20]. Among them, group fairness notions are one of the most popular [21–23], which require ML models to perform similarly for different demographic groups. In this paper, we mainly focus on the two most widely-used group fairness definitions, demographic parity [21] and equalized odds [22], but it is worth mentioning that our method is general for other fairness notions. Existing fairness promoting methods can be broadly categorized into pre-processing, in-processing, and post-processing methods [24]. Pre-processing methods calibrate the training data to remove the spurious correlations and train fair model on the modified data [25–28, 2, 1, 29, 30]. In-processing methods work on training ML with extra fairness-aware regularization [3–7, 31, 32]. For example, an adversarial framework is introduced to train model parameters to meet fairness requirements [33]. In our method, we adopt a similar adversarial loss but optimize the fairness triggers with a fixed model. Despite the effectiveness, these methods usually consider training fair models from scratch and do not directly apply to already-trained models. Post-processing methods focus on calibrating trained ML models to be fair [24]. Many of them modify the model outputs to meet the fairness criteria [18, 22, 34–44]. For example, the model outputs are directly modified to meet equalized odds by solving an optimization problem [22]. Alternatively, a boosting-based method is introduced to calibrate model outputs [40].

**Model reprogramming**   Model reprogramming [45, 9, 8, 46–48] aims to repurpose an already trained neural network for different tasks. Different from the typical transfer learning that requires modifying the structure and parameters of the given pre-trained model, reprogramming technology instead designs a trainable program appended to the input, while keeping the pre-trained model intact. The model reprogramming technology can be designed in the form of an input-agnostic

perturbation [45, 8] or a trainable input transformation function together with the label mapping from the source domain to the target domain [9]. In particular, the feasibility of designing a universal input perturbation to reprogram a well-trained ImageNet classifier to the CIFAR-10 dataset is demonstrated in the white-box setting [8]. As an exploration to implement reprogramming in the discrete scenario, another work [46] successfully reprograms the text classification neural network for alternate classification tasks. This work also shows the possibility of developing reprogramming in the black-box setting, where the reprogrammer may not have the access to the parameters of the target model. Recent work [47] shows the possibility of repurposing deep neural networks designed for image classifiers for the natural language processing and other sequence classification tasks. It is argued the success of the reprogramming lies in the size of the average input gradient and the input dimension is crucial to the performance of the reprogrammer [48]. It is also shown that generative models like FairGANs [49] can be transfered to other tasks by reprogramming with variational auto-encoders [50]. A highly related topic to model reprograming is prompt learning in NLP [11]. It is shown that by designing designated text prompts appended to inputs, pre-trained language models could be re-directed to perform well under downstream tasks in a few-shot setting [51]. Prompt-based tuning methods have become the mainstream and achieve better performance than fine-tuning in many scenarios [52-55]. Seminal works about prompt learning can be found in [11, 12]. *However*, nearly all existing methods focus on using model reprogramming to improve accuracy in domain-transfer tasks and to our best knowledge, our work is the first to generalize model reprogramming to improve fairness of a trained model.

## 3 Fairness Reprogramming

In this section, we will introduce the FAIRREPROGRAM algorithms. As some notations, upper-cased letters, $X$ and $X$, denote random vectors and random variables, respectively; lower-cased letters, $x$ or $x$, denote deterministic vectors and scalars respectively. $p_X(\cdot)$ or $p(X)$ denote the probability density function of the (discrete) random variable $X$.

### 3.1 Problem Formulation

Consider a classification task, where $X$ represents the input feature, and $Y$ represents the output label. In addition, there exists some sensitive attributes or demographic group, $Z$, that may be spuriously correlated with $Y$. There is a pre-trained classifier, $f^*(\cdot)$, that predicts $Y$ from $X$, *i.e.* $\hat{Y} = f^*(X)$. The weights of the classifier are considered fixed (hence the superscript $*$). Unfortunately, due to the spurious correlation between $Z$ and $Y$, the classifier may be biased against certain demographics.

Our goal is to improve the fairness of the classifier by modifying the input $X$, rather than modifying the classifier's fixed weights. In particular, we aim to achieve either of the following fairness criteria.

$$\textbf{Equalized Odds:} \quad \hat{Y} \perp Z | Y, \quad \text{or} \quad \textbf{Demographic Parity:} \quad \hat{Y} \perp Z, \tag{1}$$

where $\perp$ denotes independence. The following two subsection will explain how to modify input and design the optimization objective respectively.

### 3.2 Modifying the Input Features

Input modification primarily involves appending a *fairness trigger* to the input. Formally, the input modification takes the following generic form:

$$\tilde{X} = m(X; \theta, \delta) = [\delta, g(X; \theta)], \tag{2}$$

where $\tilde{X}$ denotes the modified input; $[\cdot]$ denotes vector concatenation. As can be observed, the input modification consists of two steps. First, $X$ is fed through a transformation function $g(\cdot; \theta)$, where $\theta$ represents the hyper-parameters of the transformation function. The actual form of $g(\cdot; \theta)$ is contingent upon different applications and modalities, but a general requirement is that $g(\cdot; \theta)$ should largely retain the information necessary for classification. The second step is to append a fairness trigger, $\delta$, to the input, which is a vector that can be optimized over. It is important to note that $\delta$ is a *constant* – different inputs get appended the same trigger. Although it does not seem intuitive, we will soon show that a constant trigger is all you need to achieve fair prediction on all different inputs.

Below are specific forms of transformations (Eq. (2)) we use.

**Text Classification** In text classification, $\boldsymbol{X}$ represents a sequence of input token embeddings. To modify the input, we simply append a fixed number of embeddings after the input text. In this case, $g(\cdot; \boldsymbol{\theta})$ is the identity mapping, and $\boldsymbol{\delta}$ corresponds to the appended embeddings.

**Image Classification** In image classification, $\boldsymbol{X}$ represents the (vectorized) input image. Unlike text classification, where the input can have a variable length, the length of the input to the image classification network is fixed. We thus apply the following two approaches to append the trigger, as shown in Fig. 1. The first approach, called the *patch approach*, removes a patch from the original image, and appends a trigger the same size as the patch to the patch location (as shown in Fig. 1(a)). In this case, $g(\cdot; \boldsymbol{\theta})$ is a function that removes the patch dimension and retain the rest, with $\boldsymbol{\theta}$ representing the patch location; $\boldsymbol{\delta}$ represents the trigger feature that replaces the patch. The second approach, called the *border approach*, shrinks the image to a

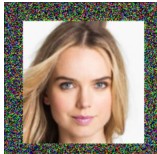 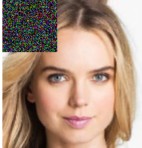

(a) Border trigger    (b) Patch trigger

Figure 1: Demonstration of the border and patch trigger applied on an image from CelebA [56].

smaller image, and then appends the trigger at the border (as shown in Fig. 1(b)). In this case, $g(\cdot; \boldsymbol{\theta})$ is a function that shrinks the image, and $\boldsymbol{\delta}$ represents the trigger feature at the border.

## 3.3 Optimization Objective

Our optimization objective is as follows

$$\min_{\boldsymbol{\delta}, \boldsymbol{\theta}} \mathcal{L}_{util}(\mathcal{D}_{tune}, f^* \circ m) + \lambda \mathcal{L}_{fair}(\mathcal{D}_{tune}, f^* \circ m), \tag{3}$$

where $m = m(\cdot; \boldsymbol{\theta}, \boldsymbol{\delta})$ represents the input modification function as in Eq. (2); $\circ$ represents nested functions; $\mathcal{D}_{tune}$ represents the dataset that are used to train the fairness trigger. Note that this is different from the dataset where the classifier, $f^*$, is pre-trained.

The first loss term, $\mathcal{L}_{util}$, is the utility loss function of the task. For classification tasks, $\mathcal{L}_{util}$ is usually the cross-entropy loss, *i.e.*,

$$\mathcal{L}_{util}(\mathcal{D}_{tune}, f^* \circ m) = \mathbb{E}_{\boldsymbol{X}, Y \sim \mathcal{D}_{tune}}[\text{CE}(Y, f^*(m(\boldsymbol{X})))], \tag{4}$$

where $\text{CE}(\cdot, \cdot)$ denotes the cross-entropy loss.

The second loss term, $\mathcal{L}_{fair}$, encourages the prediction to follow the fairness criteria as in Eq. (1). According to Eq. (1), $\mathcal{L}_{fair}$ should measure how much information about $Z$ is in $\hat{Y}$. To measure this, we introduce another network, called the discriminator, $d(\cdot; \boldsymbol{\phi})$, where $\boldsymbol{\phi}$ represents its parameters. If the equalized odds criterion is applied, then $d(\cdot; \boldsymbol{\phi})$ should predict $Z$ from $\hat{Y}$ and $Y$; if the demographic parity criterion is applied, then the input to $d(\cdot; \boldsymbol{\phi})$ would just be $\hat{Y}$. In the following, we will focus on equalize odds criterion for conciseness. Then, the information of $Z$ can be measured by maximizing the *negative* cross-entropy loss for the prediction of $Z$ over the discriminator parameters, *i.e.*,

$$\mathcal{L}_{fair}(\mathcal{D}_{tune}, f^* \circ m) = \max_{\boldsymbol{\phi}} \mathbb{E}_{\boldsymbol{X}, Y, Z \sim \mathcal{D}_{tune}}[-\text{CE}(Z, d(f^*(m(\boldsymbol{X})), Y; \boldsymbol{\phi}))]. \tag{5}$$

By plugging Eqs. (4) and (5) into (3), we can see that the entire optimization objective becomes a min-max framework, where the discriminator tries to improve its prediction of $Z$ while the fairness trigger tries to make the prediction worse. As shown in [33], when the discriminator cannot predict $Z$ better than chance, the aforementioned fairness criteria can be achieved.

## 3.4 Why Does It Work?

It is not immediately straightforward why a *global* trigger can obscure the demographic information for *any* input. In this section, we will propose an information-theoretic framework that illustrates one of the mechanisms through which the trigger can remove the demographic information.

Our theoretical framework builds upon the data generation process as shown in Fig. 2(a). Specifically, we assume that $\boldsymbol{X}$ consists of a set of features, *i.e.* $\boldsymbol{X} = [\boldsymbol{X}_1, \cdots, \boldsymbol{X}_T]$, where $T$ is the total number of features. In text classification, a feature can be a word or a word piece; in image classification, a feature can be specific shapes, colors, patterns, *etc*. Assume that these features can be divided into two groups. The first group, denoted as $\boldsymbol{X}^{(y)}$, consists of features that are directly governed by the output label $Y$; the second group, denoted as $\boldsymbol{X}^{(z)}$, consists of featuers that are directly governed by

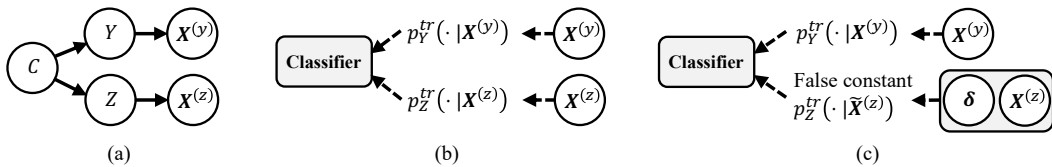

(a)             (b)             (c)

Figure 2: Illustration of why fairness trigger works. (a) The data generation process. (b) The information flow from data to the classifier through the sufficient statistics. (c) Fairness trigger strongly indicative of a demographic group can confuse the classifier with a false demographic posterior, and thus preventing the classifier from using the correct demographic information.

the demographic information $Z$. $Z$ and $Y$ can be spuriously correlated, *i.e.* there can be common confounders, $C$, between $Z$ and $Y$. As a result, both $\boldsymbol{X}^{(y)}$ and $\boldsymbol{X}^{(z)}$ are indicative of $Y$.

To further simplify our theoretical analysis, we consider a bag-of-feature scenario, where each feature in $\boldsymbol{X}^{(y)}$ is drawn from the vocabulary set $\mathcal{X}^{(y)}$, and each feature in $\boldsymbol{X}^{(z)}$ is drawn from the vocabulary set $\mathcal{X}^{(z)}$. There should not be any overlap between the two vocabulary sets, *i.e.* $\mathcal{X}^{(y)} \cap \mathcal{X}^{(z)} = \varnothing$. Otherwise it violates our assumption that demographic-related features are biased features.

It can be shown (in Appendix C) that the posterior distributions, $p_Y(\cdot|\boldsymbol{X}^{(y)})$ and $p_Z(\cdot|\boldsymbol{X}^{(z)})$, are the sufficient statistics of $\boldsymbol{X}^{(y)}$ and $\boldsymbol{X}^{(z)}$ respectively for inferring $Y$. In other words, these two posterior distributions summarize all the information about $\boldsymbol{X}^{(y)}$ and $\boldsymbol{X}^{(z)}$ that the classifier needs to know to predict $Y$. Therefore, we assume that the classifier takes the following generic form

$$\hat{Y} = f^*(\boldsymbol{X}) = h(p_Y^{tr}(\cdot|\boldsymbol{X}^{(y)}), p_Z^{tr}(\cdot|\boldsymbol{X}^{(z)})). \tag{6}$$

Note that we add a superscript, $tr$, to emphasize that the probability distributions are over the data set where the classifier is trained, because the classifier has never been trained on inputs modified with the fairness trigger. Eq. (6) encompasses many common decision functions. For example, it can be shown (in Appendix C) that the posterior distribution $p(Y|\boldsymbol{X})$, which is the minimizer of the cross-entropy loss, is a special case of Eq. (6).

As illustrated in Fig. 2(b), $p_Y(\cdot|\boldsymbol{X}^{(y)})$ and $p_Z(\cdot|\boldsymbol{X}^{(z)})$ provide two sets of information from input features. $p_Y(\cdot|\boldsymbol{X}^{(y)})$ provides the *unbiased* information, because a desirable fair classifier should rely only upon $p_Y(\cdot|\boldsymbol{X}^{(y)})$ to make a decision. On the other hand, $p_Z(\cdot|\boldsymbol{X}^{(z)})$ provides the *biased* information, because it conveys the demographic information. In other words, the fairness goals can be achieved by cutting off the biased information path. Therefore, our research question boils down to: is it possible to cut off the biased information path with a global fairness trigger $\boldsymbol{\delta}$?

Without loss of generality, assume that $\boldsymbol{\delta}$ consists of only one feature. Consider the case where $\boldsymbol{\delta}$ is a demographic feature, *i.e.* $\boldsymbol{\delta} \in \mathcal{X}^{(z)}$. In this case, we assume the transformed input as defined in Eq. (2) can also be divided into two groups:

$$\tilde{\boldsymbol{X}} = [\tilde{\boldsymbol{X}}^{(y)}, \tilde{\boldsymbol{X}}^{(z)}], \quad \text{where} \quad \tilde{\boldsymbol{X}}^{(y)} = g(\boldsymbol{X}^{(y)}), \quad \tilde{\boldsymbol{X}}^{(z)} = [\boldsymbol{\delta}, g(\boldsymbol{X}^{(z)})]. \tag{7}$$

The following theorem states our main conclusion:

**Theorem 1.** *Under the assumptions in Eq. (6) and (7), and some additional regularity conditions[2], if the fairness trigger $\boldsymbol{\delta}$ is indicative of a certain demographic group $z$, then*

$$\lim_{p^{tr}(Z=z|\boldsymbol{X}_0^{(z)}=\boldsymbol{\delta})\to 1} MI(\hat{\tilde{Y}}, Z|Y) = 0, \tag{8}$$

*where MI means mutual information; $\hat{\tilde{Y}} = f^*(\tilde{\boldsymbol{X}})$ is the classifier's prediction after input is modified.*

$p^{tr}(Z=z|\boldsymbol{X}_0^{(z)} = \boldsymbol{\delta}) \to 1$ means that the fairness trigger is very strongly indicative of the demographic group $z$. Therefore, Thm. 1 essentially states that if the prepended trigger feature is very strongly indicative of a certain demographic group, then equalized odds can be achieved. A formal proof is presented in Appendix C. Here we would like to give an intuitive explanation. When $p^{tr}(Z=z|\boldsymbol{X}_0^{(z)} = \boldsymbol{\delta}) \to 1$, it will also happen that $p^{tr}(Z=z|\boldsymbol{X}^{(z)} = \tilde{\boldsymbol{X}}^{(z)}) \to 1$. In other words, the fairness trigger $\boldsymbol{\delta}$ would overshadow the rest of the demographic features and 'trick' the classifier into believing all the

---

[2]Formal assumptions stated in the appendix.

different inputs belong to the same demographic group $z$. As a result, the second argument in Eq (6) would reduce to a constant (1 for demographic group $z$ and 0 elsewhere), effectively blocking the biased information path, as shown in Fig. 2(c). Note that the premise for the fairness trigger to work is that the classifier has never seen the modified input. Otherwise, the classifier will be able to learn to ignore the constant trigger and still elicit the true demographic information from input.

## 4 Experiments

In this section, we evaluate the effectiveness of FAIRREPROGRAM on both NLP and CV applications in terms of accuracy, fairness, performances under low-data regime, transferability and interpretability.

### 4.1 Experiment Setup

**Datasets**   We consider the following two commonly used NLP and CV datasets:
- `Civil Comments` [57, 58]: The dataset contains 448k texts with labels that depict the toxicity of each input. The demographic information of each text is provided.
- `CelebA` [56]: The dataset contains over 200k human face images and each contains 39 binary attribute annotations. We follow the conventional setting [56] that adopts the hair color prediction task in our experiment and uses gender annotation as the demographic information. [59–61]

For both datasets, we split the entire data into a training set, a tuning set, a validation set, and a testing set. The training set is used for the base model training, *i.e.*, to obtain a biased model for reprogramming. The tunning set and validation set are used for trigger training and hyper-parameter selection. We report our results on the testing set. It is worth mentioning that there is no overlapping data between different sets and the size of the tuning set is much smaller than the training one. Specifically, we set the size ratio between the tunning set and the training as $1/5$ and $1/100$ for `Civil Comments` and `CelebA`, respectively. The full statistics of the datasets can be found in Appendix A.1.

**Metrics**   Besides the model accuracy, we introduce two empirical fairness metrics, one under each of the two fairness criteria as in Eq. (1). For binary classification, the metrics are calculated as:

$$\textbf{DP:} \sum_{z \in \mathcal{Z}} |p(\hat{Y} = 1) - p(\hat{Y} = 1|Z = z)|, \quad \textbf{EO:} \sum_{z \in \mathcal{Z}} (|\text{FPR} - \text{FPR}_z| + |\text{FNR} - \text{FNR}_z|)/2,$$

where DP and EO stand for demographic parity and equalized odds respectively. FPR and FNR are the false positive/negative rate, and the subscript $z$ denotes the score is calculated within a specific demographic group $Z = z$. For example, $\text{FPR}_{male}$ indicates the false positive rate calculated over all examples with the "male" annotation. For a multi-class setting, the bias scores are first calculated similarly using one-vs-all for each class and then averaged across different classes. All reported results are the average of three different random runs. It can be shown that these metrics are non-negative, and will become zero when their corresponding fairness criteria are achieved. For better elaboration, we report the negative bias scores in our experiments, so the larger these negative scores are, the better the model satisfies the corresponding fairness criteria.

**Baselines and implementation details**   We consider the following models for comparison:

• BASE: the base model to be reprogrammed, trained with the cross-entropy loss on the training set.

• ADVIN [33]: an in-processing adversarial training method that optimizes both model accuracy and fairness using the training set.

• ADVPOST: a post-processing variant of ADVIN, which fine-tunes the BASE model with the same fairness-aware adversarial objectives as ADVIN, but using the (low-resource) tunning set only.

For NLP experiments, we use a pre-trained BERT [62] to obtain the BASE and ADVIN models. We use ADAMW [63] as the optimizer, and set the learning rate to $10^{-5}$ for all baselines and $0.1$ for FAIRREPROGRAM. For CV experiments, we consider a RESNET-18 [64] that pre-trained on ImageNet. The discriminator used in ADVIN, ADVPOST and FAIRREPROGRAM is a three-layer MLP, and the parameters are optimized using ADAM with a learning rate of $0.01$. We pick the best model based on the accuracy (for the BASE) or the bias scores (for all other debiasing methods) of the validation set. We refer to Appendix A.2 for more details and Appendix B for more baseline studies.

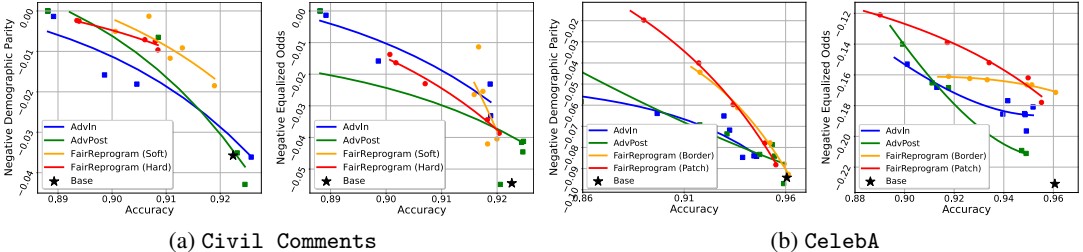

(a) `Civil Comments`        (b) `CelebA`

Figure 3: Results on (a) `Civil Comments` and (b) `CelebA`. We report the negative DP (left) and the negative EO (right) scores. For each method, we vary the trade-off parameter $\lambda$ (as shown in (3)) to record the performance. The closer a dot to the upper-right corner, the better the model is. We consider five different $\lambda$s for each method. The solid curve is the fitted polynomial with order 30.

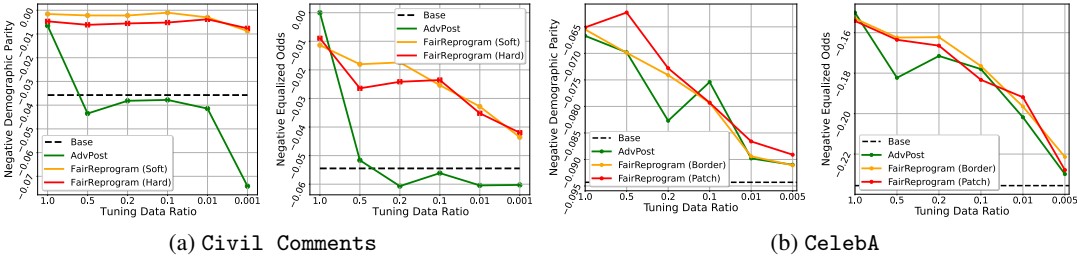

(a) `Civil Comments`        (b) `CelebA`

Figure 4: Results on (a) `Civil Comments` and (b) `CelebA` with different tuning data ratio. We report the negative DP (left) and negative EO (right) scores. We consider a fixed BASE model trained with training set, whose negative bias scores are presented as a black dashed line. Then we train other methods with different tuning data ratio to promote fairness of the BASE model.

Next we introduce the implementation details of the triggers for different variants of FAIRREPRO-GRAM. For image classification task, we adopt the border and patch trigger as shown in Fig. 1, termed FAIRREPROGRAM (BORDER) and FAIRREPROGRAM (PATCH) correspondingly. We define the trigger size as the width of the trigger frame for border trigger and the width of the square patch for patch trigger. Unless otherwise stated, the default trigger size for each setting are 20 and 80.

For text classification task, we introduce a probability vector $v_i$ to control the selection of trigger word for each position $i$. Specifically, we have the trigger $\delta_i = Ev_i$ where $E$ represents the pretrained word embedding matrix of BERT. Then we simply concatenate $\delta$ after all input texts[3] in the embeddings space as the fairness trigger. We introduce two types of trigger. The first type, called FAIRREPROGRAM (SOFT) , uses continuous $v_i$'s, and each $v_i$ is projected onto the continuous probability simplex using the bisection algorithm after each training step. The second type, called FAIRREPROGRAM (HARD), discretizes each $v_i$ into a one-hot vector $\hat{v}_i$ via $\arg\max$ operation. We adopt the straight through technique [65] to update $v_i$ during training. The triggers found by FAIRREPROGRAM (HARD) enjoy better interpretability as they correspond to a sequence of word tokens. Unless specified otherwise, we set the trigger word number as five for our experiments.

### 4.2 Results

Fig. 3 shows the performance of the proposed FAIRREPROGRAM with other baselines on both NLP (subfigure (a)) and CV (subfigure (b)) datasets using DP (left) and EO (right) metrics. In each subfigure, the data samples of the same method (dots in the same color) are generated by explicit changing the adversary weight $\lambda$ in (3), which controls the trade-off between fairness and accuracy. We further fit the data with polynomial regression to present the curves. Appendix A.2 shows the detailed $\lambda$ choices for different methods. Here are our key observations. First, our method improves the fairness of the BASE model. In particular, our methods (both orange and red curves) achieve higher negative DP and EO scores with a comparable classification accuracy. Second, our method enjoys a better fairness-accuracy trade-off compared with all other baselines. Specifically, the curves of our method lie farther to the upper-right corner of the plots, which implies that our method

---

[3]The trigger is appended as a suffix after all input tokens but before [SEP] for BERT.

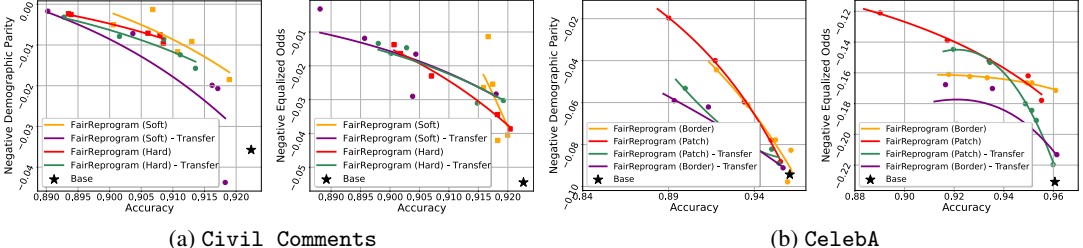

(a) `Civil Comments`                                    (b) `CelebA`

Figure 5: Results in the transfer setting. We report negative DP (left) and negative EO (right) scores. The triggers are firstly trained in a BASE model. Then we evaluate the triggers based on another unseen BASE model. We change the parameter $\lambda$ to trade-off accuracy with fairness and draw the curves in the same way with Fig. 3. The ★ point corresponds to the average of all BASE models with different random seeds.

improves model fairness with fewer sacrifices on accuracy. It is also worth noting that although ADVIN achieves good fairness scores, it uses much more data for training.

**Limited data setting** We further evaluate ADVPOST and FAIRREPROGRAM with decreasing the number of data in the tuning set. Specifically, we fix a $\lambda$ for each method such that all methods achieve comparable bias score with full tuning set. The detailed $\lambda$ choices are provided in Appendix A.2. Then we apply these methods to subsets of the tuning set with different proportions. The results are shown in Fig. 4. There are two key observations. First, our method can consistently improve fairness upon BASE model even with 1% tuning data, indicating a high data efficiency of FAIRREPROGRAM. Second, FAIRREPROGRAM achieves better fairness than ADVPOST does when tuning data number decreases. For example, in Fig. 4 (a), the curve of our method is significantly above the ADVPOST as tuning data decreases. When the tuning set size is extremely small, ADVPOST significantly deteriorates and even underperforms the BASE model.

**Transferability** Next, we show the transferability of the fairness triggers found by FAIRREPROGRAM. We first tune the triggers with a BASE source model and then apply the trigger on a target model trained with a different random seed. The results are shown in Fig. 5. As can be seen, FAIRRE-PROGRAM achieves comparable fairness-accuracy trigger on both the source model and the target model, indicating our method has a good transferability. This intriguing property brings two benefits of our method: ① if ML model parameters are infeasible (*e.g.* when ML models are provided as services), the users could train a surrogate model and tune the trigger based on it to promote fairness of the original model; ② when ML model parameters are updated with new data (*e.g.* online learning), the user could still use the original trigger for fixing fairness problems. We further elaborate the results of FAIRREPROGRAM for transferring to different tasks and model architectures in Appendix B.5.

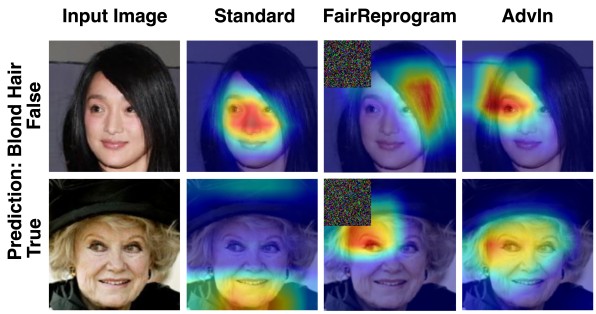

Figure 6: Gradient-based saliency map visualized with GRAD CAM [66] of different methods. The highlighted zones (marked in red) depicting regions exerting major influence on the predicted labels (non-blond hair v.s. blond hair) in each row, which also depict the attention of the model on the input image.

**Input saliency attribution.** Fig. 7 and 6 compare the saliency maps of some example inputs with and without the fairness triggers. Specifically, For the NLP applications, we extract a subset of `Civil Comments` with religion-related demographic annotations, and apply IG [67] to localize word pieces that contribute most to the text toxicity classification. For the CV application, we use GradCam [66] to identify class-discriminative regions of `CelebA`'s test images. As shown in Fig. 7, our fairness trigger consists of a lot of religion-related words (*e.g.*, diocesan, hebrew, parish). Meanwhile, the predicted toxicity score of the benign text starting from 'muslims' significantly reduces. These observations verify our theoretical hypothesis that the fairness trigger is strongly indicative of a certain demographic group to prevent the classifier from using the true demographic information. In addition, Fig. 6 presents the input saliency maps on two input images with respect to their predicted

| Text (Non-toxic) | | Predicted Toxicity |
|---|---|---|
| **muslims** need to take a look in **the** **mirror** | | 0.149 |
| **muslims** need to take a look in the mirror | same **diocesan** **bula** **rev** **proceedings** | 0.069 |
| **muslims** need to take a look in the mirror | soto **cc** rib **hebrew** armenian | 0.054 |
| **muslims** need to take a look in the mirror | **paul** long course **parish** **body** | 0.073 |

Figure 7: A text example from `Civil Comments` with INTEGRATED GRADIENT [67, 68] highlighting important words that influence ERM model predictions. The text is concatenated with three triggers generated with different adversary weight. **Green highlights** the words that lean to toxic predictions and **Red highlights** non-toxic leaning words. The model prediction tends to be correct after adding the triggers.

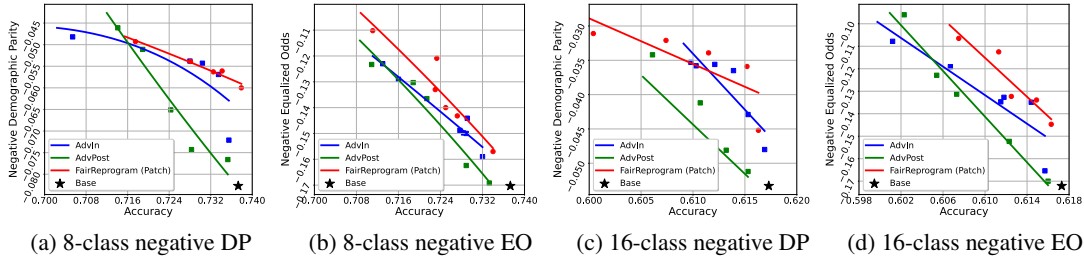

(a) 8-class negative DP     (b) 8-class negative EO     (c) 16-class negative DP     (d) 16-class negative EO

Figure 8: Performance of multi-class classification. For (a) and (b), we use the attributes *Blond Hair, Smiling, Attractive* for multi-class construction. We add an addition attribute *Wavy Hair* for (c) and (d).

labels, non-blond hair and blond hair, respectively. As can be observed, when there is no fairness trigger, the saliency region incorrectly concentrates on the facial parts, indicating the classifier is likely to use biased information, such as gender, for its decision. With the fairness trigger, the saliency region moves to the hair parts, which matches the behavior of ADVIN. These results confirm that our fairness trigger can drive models to make fairer predictions.

To further verify that the triggers encode demographic information, we trained a demographic classifier to predict the demographics from the input (texts or images) without triggers. The obtained demographic classifiers can accurately identify the demographics contained in the inputs and achieve over 0.99 AUC for identifying demographics in the validation datasets. Then, we use the demographic classifier to predict the demographic information of a null image/text[4] with the trigger. Specifically, we select three triggers generated with different $\lambda$ values for both two datasets. The results[5] can be seen in Table 1. We see that the demographic classifier gives confident outputs on the triggers. For example, we see that the trigger *paul long course parish body* is classified as containing *christian* with 0.98 confidence, indicating that the found triggers are highly indicative of demographics. This is consistent with our perspective in Section 3.4 that the fairness triggers are encoding fake demographic information to obscure ML models from making biased predictions.

Table 1: Predictions of the demographic classifier on a null input with triggers generated by different $\lambda$. The demographic prediction for CV triggers indicate the predicted score for *Male* and *Female*, and it is *Christian*, *Muslim* and *Other religion* for NLP.

| Trigger | Demographic Prediction |
|---|---|
| | 0.85, 0.15 |
| | 0.92, 0.08 |
| | 0.80, 0.20 |
| same diocesan bula rev proceedings | 0.96, 0.11, 0.02 |
| soto cc rib hebrew armenian | 0.51, 0.08, 0.81 |
| paul long course parish body | 0.98, 0.04, 0.03 |

### 4.3 Multi-Class Classification

To extend our evaluation to a multi-class setting, we use the `CelebA` dataset and select $n$ binary attributes that may be spuriously correlated with *gender* [59–61]. Then, following [69], we construct data groups by enumerating all $2^n$ possible binary vectors, where each dimension corresponds to a binary attribute. We index these vectors and treat them as the class labels. Fig. 8 shows the accuracy-fairness trade-off curves similar to Fig. 3. It can be observed that our method outperforms the other methods as the red curves are closer to the top-right corner. Also, as the class label number

---

[4]We use an empty string as the null text and an all-black image as the null image.

[5]One text could contain multiple religions so the probabilities do not sum to one for NLP triggers.

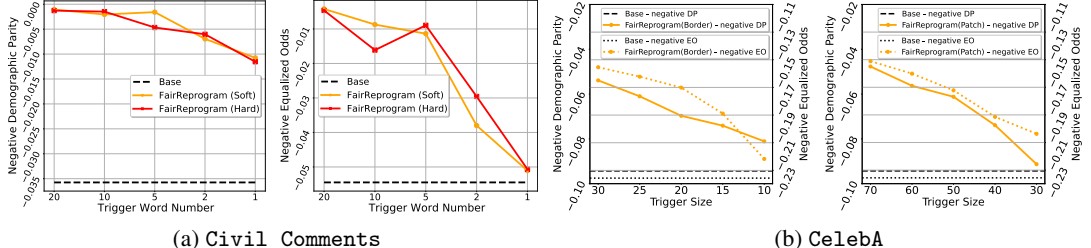

(a) `Civil Comments`     (b) `CelebA`

Figure 9: Ablation study of the trigger size. We evaluate the bias scores with different trigger word numbers (`Civil Comments`) and different trigger size (`CelebA`) with fixed adversary weight $\lambda$.

increases, the post-processing-based ADVPOST falls behind its in-processing counterpart ADVIN, indicating a larger class number may induce more challenges to post-processing methods.

### 4.4 Ablation Studies

We perform an ablation study to investigate the effects of the trigger size. Specifically, we run experiments with different numbers of trigger words / trigger patch sizes on the NLP / CV dataset. We set a $\lambda$ value for each method such that all methods achieve comparable bias scores with the largest trigger size. The detailed $\lambda$ choices can be seen in Appendix A.2. Then we train the triggers with different sizes in the tuning set using the fixed $\lambda$'s. For the text trigger as shown in Fig. 9(a), we see that the negative bias score gets worse as the number of trigger words gets smaller. However, our method can still improve fairness upon the BASE model even with only a one-word trigger. On the other hand, the results with five trigger words and above are all comparable, indicating that five words is enough to achieve the fairness goal. Similarly, for the image trigger as shown in Fig. 9(b), the results suggest a larger trigger would consistently improve fairness. On the other hand, we show that larger trigger size could hurt accuracy in Appendix B, which is similar to the effect of increasing $\lambda$.

### 4.5 Summary of Additional Results

We compare our proposed FAIRREPROGRAM with four additional baselines, and we show the full results with variance in Tab. 3. We further compare our method with MMD methods where $\mathcal{L}_{fair}$ in Eq. (3) is replaced with Maximum Mean Discrepancy regularization [70] to partial out the instability of adversarial training, and the results are shown in Fig. 10. We also implement the fairness reprogramming in the black-box setting on `CelebA` dataset, where the model parameters are not available for training the reprogram, and the results are shown in Fig. 11. Besides, we show that FAIRREPROGRAM could also be used in tabular data, and the corresponding experiment results on the `Adult` dataset are shown in Fig. 13.

## 5 Conclusion

In this paper, we introduce a novel model reprogramming based fairness promoting method, termed FAIRREPROGRAM. Specifically, FAIRREPROGRAM considers a fixed ML model and optimizes a set of vectors, named fairness trigger, concatenated on inputs to boost model fairness. We introduce an information-theoretic framework to explain the rationales of why FAIRREPROGRAM can improve model fairness. As implied by our theoretic framework as well as our empirical findings, the fairness trigger can effectively mask out the true demographic information with its strong, false demographic information. Extensive experiments demonstrate that our method could achieve better fairness improvements to retraining based methods with far-less training cost. We further empirically show fairness triggers enjoys great transferability and interpretability. We hope that FAIRREPROGRAM can inspire new fairness learning paradigms that are more feasible and flexible in practice.

### Acknowledgement

The work of Yihua Zhang, Sijia Liu, and Shiyu Chang was partially supported by National Science Foundation (NSF) Grant IIS-2207052. The computing resources used in this work were partially supported by the MIT-IBM Watson AI Lab.

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
