Table 2: Statistics of the datasets. "Pos. (%)" column indicates the ratio of positive labels (*i.e.* "blond hair" for CelebA, "toxic" for Civil Comments). We use "Tr.", "Tun.", "Val." and "test." columns to indicate the size of training set, tuning set, validation set and testing set. "Demographics" column indicates the considered demographics in each dataset.

| Dataset | Task | Pos. (%) | Tr. | Tun. | Val. | Test. | Demographics |
|---|---|---|---|---|---|---|---|
| CelebA | Hair color recognition | 17.4 | 161143 | 1627 | 19867 | 19962 | Gender |
| Civil Comments | Toxicity classification | 11.3 | 223858 | 45180 | 45180 | 133782 | Gender, Sex orientation, Race, Religion |

# A  Experiment Setup

## A.1  Dataset Details

The dataset splitting setting and demographic information of the datasets are shown in Tab. 2.

## A.2  Training Details

We specify the different $\lambda$ values used to generate the curves in Figs. 3 and Figs. 5 in Tab. 3.

For Civil Comments in Figs 4 and 9, we set $\lambda = 0.5$ for ADVPOST, $\lambda = 50.0$ for FAIRREPROGRAM (SOFT) and $\lambda = 1000.0$ for FAIRREPROGRAM (HARD) with the DP measure; we set $\lambda = 1.0$ for ADVPOST and $\lambda = 50.0$ for both of our methods with the EO measure.

For CelebA in Fig. 4 and Fig. 9, we set $\lambda = 1.0$ for FAIRREPROGRAM (BORDER) and FAIRRE-PROGRAM (PATCH) with DP and we set $\lambda = 10.0$ for both with EO. By default, the trigger size of FAIRREPROGRAM (BORDER) is set to 20, which corresponds to the width of the trigger frame. The trigger size of FAIRREPROGRAM (PATCH) is fixed to 80, namely the width of the trigger block attached to the original input image. For ADVIN and ADVPOST, the $\lambda$ is set to 0.1 in the setting with DP and $\lambda$ is fixed to 0.5 for training with EO. The value of $\lambda$ is selected so that different methods achieve comparable bias scores.

## B  Additional Experiment Results

### B.1  Experiments with Additional Post-processing Baselines

Table 3: Numerical results with standard derivation on `Civil Comments` and `CelebA` shown in Fig. 3. All reported results are the average of three different random runs. We report the negative DP and the negative EO scores correspondingly for the "Fairness" column. Note that the best models are also selected based on corresponding fairness measures.

| Method | Civil Comments | | | | | | CelebA | | | | | |
|---|---|---|---|---|---|---|---|---|---|---|---|---|
| | Demographic parity | | | Equalized odds | | | Demographic parity | | | Equalized odds | | |
| | $\lambda$ | Accuracy | Fairness | $\lambda$ | Accuracy | Fairness | $\lambda$ | Accuracy | Fairness | $\lambda$ | Accuracy | Fairness |
| ERM | - | $0.922_{\pm 0.004}$ | $-0.036_{\pm 0.005}$ | - | $0.923_{\pm 0.004}$ | $-0.054_{\pm 0.025}$ | - | $0.961_{\pm 0.004}$ | $-0.094_{\pm 0.002}$ | - | $0.961_{\pm 0.004}$ | $-0.231_{\pm 0.008}$ |
| ADVIN | 0.0 | $0.926_{\pm 0.004}$ | $-0.036_{\pm 0.025}$ | 0.0 | $0.919_{\pm 0.004}$ | $-0.033_{\pm 0.005}$ | 0.01 | $0.944_{\pm 0.007}$ | $-0.084_{\pm 0.004}$ | 0.1 | $0.952_{\pm 0.005}$ | $-0.181_{\pm 0.009}$ |
| | 0.1 | $0.899_{\pm 0.014}$ | $-0.016_{\pm 0.004}$ | 0.1 | $0.899_{\pm 0.006}$ | $-0.016_{\pm 0.009}$ | 0.05 | $0.938_{\pm 0.005}$ | $-0.085_{\pm 0.007}$ | 0.3 | $0.941_{\pm 0.002}$ | $-0.177_{\pm 0.014}$ |
| | 0.5 | $0.905_{\pm 0.003}$ | $-0.018_{\pm 0.035}$ | 1.0 | $0.919_{\pm 0.006}$ | $-0.023_{\pm 0.002}$ | 0.1 | $0.932_{\pm 0.006}$ | $-0.072_{\pm 0.005}$ | 0.5 | $0.936_{\pm 0.005}$ | $-0.175_{\pm 0.012}$ |
| | 5.0 | $0.889_{\pm 0.007}$ | $-0.001_{\pm 0.039}$ | 5.0 | $0.889_{\pm 0.005}$ | $-0.001_{\pm 0.007}$ | 0.2 | $0.911_{\pm 0.003}$ | $-0.071_{\pm 0.009}$ | 1.0 | $0.913_{\pm 0.007}$ | $-0.168_{\pm 0.009}$ |
| | 20.0 | $0.888_{\pm 0.023}$ | $-0.000_{\pm 0.039}$ | 20.0 | $0.888_{\pm 0.009}$ | $-0.000_{\pm 0.006}$ | 0.3 | $0.897_{\pm 0.002}$ | $-0.064_{\pm 0.002}$ | 2.0 | $0.901_{\pm 0.004}$ | $-0.153_{\pm 0.007}$ |
| ADVPOST | 0.0 | $0.923_{\pm 0.003}$ | $-0.035_{\pm 0.019}$ | 0.0 | $0.921_{\pm 0.005}$ | $-0.055_{\pm 0.004}$ | 0.01 | $0.959_{\pm 0.004}$ | $-0.097_{\pm 0.004}$ | 0.1 | $0.947_{\pm 0.003}$ | $-0209_{\pm 0.007}$ |
| | 0.1 | $0.925_{\pm 0.002}$ | $-0.043_{\pm 0.011}$ | 0.2 | $0.925_{\pm 0.002}$ | $-0.045_{\pm 0.002}$ | 0.05 | $0.947_{\pm 0.009}$ | $-0.083_{\pm 0.005}$ | 0.3 | $0.931_{\pm 0.005}$ | $-0.201_{\pm 0.011}$ |
| | 0.5 | $0.909_{\pm 0.011}$ | $-0.007_{\pm 0.032}$ | 0.4 | $0.925_{\pm 0.002}$ | $-0.041_{\pm 0.004}$ | 0.1 | $0.931_{\pm 0.009}$ | $-0.074_{\pm 0.007}$ | 0.5 | $0.918_{\pm 0.004}$ | $-0.168_{\pm 0.015}$ |
| | 1.0 | $0.888_{\pm 0.022}$ | $-0.000_{\pm 0.033}$ | 0.7 | $0.924_{\pm 0.002}$ | $-0.042_{\pm 0.005}$ | 0.2 | $0.917_{\pm 0.003}$ | $-0.069_{\pm 0.005}$ | 1.0 | $0.911_{\pm 0.008}$ | $-0.165_{\pm 0.011}$ |
| | 5.0 | $0.888_{\pm 0.022}$ | $-0.000_{\pm 0.033}$ | 1.0 | $0.888_{\pm 0.022}$ | $-0.000_{\pm 0.028}$ | 0.3 | $0.873_{\pm 0.002}$ | $-0.058_{\pm 0.002}$ | 2.0 | $0.899_{\pm 0.002}$ | $-0.141_{\pm 0.013}$ |
| EQODDS | - | $0.913_{\pm 0.005}$ | $-0.032_{\pm 0.020}$ | - | $0.915_{\pm 0.003}$ | $-0.031_{\pm 0.005}$ | - | $0.919_{\pm 0.009}$ | $-0.047_{\pm 0.005}$ | - | $0.919_{\pm 0.009}$ | $-0.172_{\pm 0.009}$ |
| CALIEQODDS | - | $0.922_{\pm 0.003}$ | $-0.044_{\pm 0.023}$ | - | $0.922_{\pm 0.004}$ | $-0.057_{\pm 0.011}$ | - | $0.927_{\pm 0.007}$ | $-0.053_{\pm 0.004}$ | - | $0.927_{\pm 0.007}$ | $-0.169_{\pm 0.018}$ |
| REJECTOPTION | - | $0.886_{\pm 0.028}$ | $-0.152_{\pm 0.052}$ | - | $0.874_{\pm 0.017}$ | $-0.101_{\pm 0.002}$ | - | $0.934_{\pm 0.003}$ | $-0.089_{\pm 0.004}$ | - | $0.934_{\pm 0.003}$ | $-0.189_{\pm 0.015}$ |
| DIREMOVER | - | $0.917_{\pm 0.008}$ | $-0.017_{\pm 0.017}$ | - | $0.922_{\pm 0.003}$ | $-0.034_{\pm 0.003}$ | - | $0.959_{\pm 0.004}$ | $-0.086_{\pm 0.003}$ | - | $0.959_{\pm 0.004}$ | $-0.183_{\pm 0.014}$ |
| FAIRREPROGRAM (SOFT / BORDER) | 0.0 | $0.919_{\pm 0.005}$ | $-0.018_{\pm 0.021}$ | 0.0 | $0.920_{\pm 0.004}$ | $-0.040_{\pm 0.002}$ | 0.1 | $0.961_{\pm 0.002}$ | $-0.093_{\pm 0.005}$ | 2.0 | $0.961_{\pm 0.005}$ | $-0.171_{\pm 0.005}$ |
| | 0.5 | $0.911_{\pm 0.007}$ | $-0.012_{\pm 0.018}$ | 0.1 | $0.916_{\pm 0.007}$ | $-0.026_{\pm 0.007}$ | 0.5 | $0.959_{\pm 0.005}$ | $-0.087_{\pm 0.006}$ | 5.0 | $0.951_{\pm 0.007}$ | $-0.167_{\pm 0.004}$ |
| | 5.0 | $0.913_{\pm 0.008}$ | $-0.009_{\pm 0.011}$ | 10.0 | $0.918_{\pm 0.005}$ | $-0.042_{\pm 0.006}$ | 1.0 | $0.952_{\pm 0.007}$ | $-0.078_{\pm 0.005}$ | 10.0 | $0.933_{\pm 0.003}$ | $-0.163_{\pm 0.003}$ |
| | 20.0 | $0.901_{\pm 0.014}$ | $-0.005_{\pm 0.023}$ | 20.0 | $0.917_{\pm 0.006}$ | $-0.025_{\pm 0.012}$ | 2.0 | $0.929_{\pm 0.003}$ | $-0.075_{\pm 0.004}$ | 20.0 | $0.926_{\pm 0.004}$ | $-0.162_{\pm 0.005}$ |
| | 100.0 | $0.907_{\pm 0.011}$ | $-0.001_{\pm 0.003}$ | 50.0 | $0.917_{\pm 0.004}$ | $-0.011_{\pm 0.010}$ | 5.0 | $0.911_{\pm 0.002}$ | $-0.072_{\pm 0.002}$ | 30.0 | $0.918_{\pm 0.002}$ | $-0.161_{\pm 0.003}$ |
| FAIRREPROGRAM (HARD / PATCH) | 0.0 | $0.908_{\pm 0.008}$ | $-0.010_{\pm 0.016}$ | 0.0 | $0.920_{\pm 0.004}$ | $-0.039_{\pm 0.001}$ | 0.1 | $0.955_{\pm 0.004}$ | $-0.088_{\pm 0.004}$ | 2.0 | $0.955_{\pm 0.004}$ | $-0.178_{\pm 0.011}$ |
| | 0.1 | $0.908_{\pm 0.011}$ | $-0.008_{\pm 0.022}$ | 20.0 | $0.918_{\pm 0.005}$ | $-0.034_{\pm 0.002}$ | 0.5 | $0.950_{\pm 0.005}$ | $-0.078_{\pm 0.007}$ | 5.0 | $0.946_{\pm 0.008}$ | $-0.161_{\pm 0.009}$ |
| | 10.0 | $0.906_{\pm 0.012}$ | $-0.007_{\pm 0.019}$ | 200.0 | $0.907_{\pm 0.015}$ | $-0.023_{\pm 0.017}$ | 1.0 | $0.934_{\pm 0.005}$ | $-0.060_{\pm 0.003}$ | 10.0 | $0.934_{\pm 0.004}$ | $-0.152_{\pm 0.007}$ |
| | 30.0 | $0.894_{\pm 0.017}$ | $-0.003_{\pm 0.021}$ | 600.0 | $0.902_{\pm 0.013}$ | $-0.016_{\pm 0.014}$ | 2.0 | $0.917_{\pm 0.003}$ | $-0.040_{\pm 0.008}$ | 20.0 | $0.917_{\pm 0.002}$ | $-0.139_{\pm 0.012}$ |
| | 100.0 | $0.893_{\pm 0.015}$ | $-0.002_{\pm 0.017}$ | 1200.0 | $0.901_{\pm 0.017}$ | $-0.014_{\pm 0.011}$ | 5.0 | $0.890_{\pm 0.001}$ | $-0.019_{\pm 0.002}$ | 30.0 | $0.890_{\pm 0.001}$ | $-0.121_{\pm 0.005}$ |

We further compare our method with four extra post-processing fairness-promoting baselines.

• EQODDS [22]: Method that alters model predictions to meet equalized odds by solving a linear program.

• CALIEQODDS [71]: Method that optimizes the model outputs to achieve a relaxed equalized odds objective together with calibration with information withholding.

• REJECTOPTION [27]: Method that tunes model outputs with more favorable labels to minority groups (vice versa) in the low confidence region of classifiers to achieve better demographic parity.

• DIREMOVER [72]: Disparate impact remover is proposed as a pre-processing fairness promoting method, which modifies input features with rank-ordering preserving operations. We simply apply the method to modify model predictions as a post-processing method to promote demographic parity.

EQODDS, CALIEQODDS and REJECTOPTION are trained on the tuning set and then applied on testing set while DIREMOVER directly tune the model predictions on the testing set. We use the implementation [73] for all four baselines.

The results can be seen in Table 3. We see that our method consistently outperforms these baselines with improved fairness-accuracy trade-off. For example, we see that FAIRREPROGRAM (BORDER) can achieve -0.167 negative EO and 0.951 accuracy in `CelebA` with $\lambda = 5.0$. By contrast, the best-performing post-processing baseline achieves much worse accuracy (0.927). Similar comparisons can also be seen in `Civil Comments`, where the best post-processing baseline can achieve -0.031 negative EO score and 0.915 accuracy, while our method FAIRREPROGRAM (SOFT) can achieve -0.011 negative EO with 0.917 accuracy with $\lambda = 50.0$.

### B.2  Experiments with Additional MMD Baselines

To partial out the instability of the adversarial training, we further compare our method with MMD method, where the adversarial loss $\mathcal{L}_{fair}$ in Eq. (3) is replaced with the Maximum Mean Discrepancy regularization [70]. Specifically, we consider MMDIN and MMDPOST, where model parameters are trained from scratch in an in-processing manner and fine-tuned in a post-processing manner, respectively, following the settings for adversarial training in Section 4.1. The experiment results

on the `Civil Comments` dataset are presented in Fig. 10. As we can see, our proposed method FAIRREPROGRAM outperforms the MMD baselines, which can alleviate the concern that fairness reprogramming has a better performance simply because of the instability of adversarial training of the baselines.

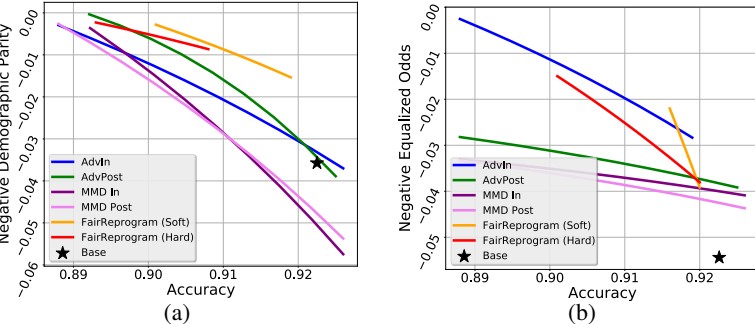

(a)            (b)

Figure 10: Results on `Civil Comments` with the new MMD baseline. We report the negative DP (left) and the negative EO (right) scores. For each method, we vary the trade-off parameter $\lambda$ (as shown in (3)) to record the performance. The closer a dot to the upper-right corner, the better the model is. We consider four different $\lambda$s for each method. The solid curve is the fitted polynomial with order 30.

## B.3 Black-box FAIRREPROGRAM Generation

Previous experiments are all based on the white-box setting, which assumes access to the complete model information, such as model architectures and parameters. This precludes the use case of reprogramming a well-trained but access-limited model, *e.g.*, a commercial APIs or other query-based software [9]. Thus, we further explore the feasibility of our method in the black-box setup [9, 74], where the gradients of the pre-trained model are estimated using only function queries. We follow the general black-box setting in [9] and adopt a query number of 30. The results are summarized in Fig. 11. As we can see, out algorithm can still improve the fairness without the knowledge of the model information. However, in such a case, the gain in fairness would sacrifice the accuracy largely when compared to our baselines. While in the future work, we will try to mitigate such degradation using more query numbers[9] and coordinate gradient estimation (CGE) [74] to achieve more accurate gradient estimation.

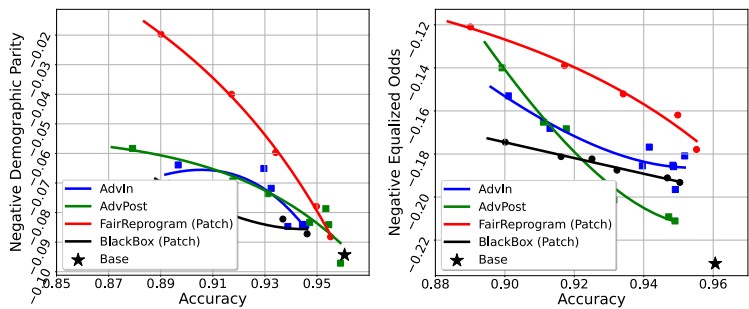

Figure 11: Performance of FAIRREPROGRAM in the black-box setting. The left Performance of the triggers trained in the black-box setting. Both the reprogrammer and the adversary are trained with query-based estimated gradients. Different data samples represent different

## B.4 Results with Standard Derivation

The numerical results in Figs. 3, 4 and 9 with standard derivation are correspondingly presented in Tabs 3, 4 and 5.

## B.5 Transfer experiments with different tasks and model architectures

We further test the transferability of reprogramming to different tasks and model architectures as shown in Fig. 12. Specifically, for transfer setting, the reprogram is optimized on the (ResNet-18,

Table 4: Numerical results with standard derivation on `Civil Comments` and `CelebA` with different tuning data ratio, corresponding to Fig. 4. All reported results are the average of three different random runs. We report the negative DP and the negative EO scores correspondingly for the "Fairness" column. We consider a fixed BASE model trained with the training set, whose negative bias scores are presented as a black dashed line. Then we train other methods with different tuning data ratios to promote fairness of the BASE model.

| Method | Tuning Data Ratio | Civil Comments | | | | CelebA | | | |
| | | Demographic parity | | Equalized odds | | Demographic parity | | Equalized odds | |
| | | Accuracy | Fairness | Accuracy | Fairness | Accuracy | Fairness | Accuracy | Fairness |
|---|---|---|---|---|---|---|---|---|---|
| ERM | - | $0.922_{\pm 0.004}$ | $-0.036_{\pm 0.025}$ | $0.923_{\pm 0.004}$ | $-0.054_{\pm 0.005}$ | $0.961_{\pm 0.004}$ | $-0.094_{\pm 0.002}$ | $0.961_{\pm 0.004}$ | $-0.231_{\pm 0.008}$ |
| ADVPOST | 1.0 | $0.909_{\pm 0.011}$ | $-0.007_{\pm 0.032}$ | $0.888_{\pm 0.022}$ | $-0.000_{\pm 0.028}$ | $0.908_{\pm 0.005}$ | $-0.067_{\pm 0.005}$ | $0.905_{\pm 0.004}$ | $-0.150_{\pm 0.007}$ |
| | 0.5 | $0.923_{\pm 0.005}$ | $-0.044_{\pm 0.009}$ | $0.919_{\pm 0.005}$ | $-0.053_{\pm 0.008}$ | $0.915_{\pm 0.007}$ | $-0.069_{\pm 0.007}$ | $0.911_{\pm 0.006}$ | $-0.162_{\pm 0.013}$ |
| | 0.2 | $0.923_{\pm 0.004}$ | $-0.038_{\pm 0.017}$ | $0.919_{\pm 0.006}$ | $-0.061_{\pm 0.004}$ | $0.939_{\pm 0.002}$ | $-0.075_{\pm 0.006}$ | $0.929_{\pm 0.005}$ | $-0.171_{\pm 0.010}$ |
| | 0.1 | $0.917_{\pm 0.007}$ | $-0.038_{\pm 0.015}$ | $0.918_{\pm 0.011}$ | $-0.056_{\pm 0.013}$ | $0.943_{\pm 0.003}$ | $-0.083_{\pm 0.004}$ | $0.933_{\pm 0.007}$ | $-0.178_{\pm 0.008}$ |
| | 0.01 | $0.922_{\pm 0.002}$ | $-0.041_{\pm 0.014}$ | $0.920_{\pm 0.010}$ | $-0.060_{\pm 0.014}$ | $0.948_{\pm 0.005}$ | $-0.089_{\pm 0.005}$ | $0.948_{\pm 0.003}$ | $-0.202_{\pm 0.012}$ |
| | 0.001 | $0.917_{\pm 0.005}$ | $-0.083_{\pm 0.018}$ | $0.921_{\pm 0.006}$ | $-0.060_{\pm 0.009}$ | $0.951_{\pm 0.007}$ | $-0.091_{\pm 0.005}$ | $0.955_{\pm 0.002}$ | $-0.229_{\pm 0.005}$ |
| FAIRREPROGRAM (SOFT/BORDER) | 1.0 | $0.917_{\pm 0.003}$ | $-0.002_{\pm 0.001}$ | $0.917_{\pm 0.004}$ | $-0.011_{\pm 0.010}$ | $0.935_{\pm 0.003}$ | $-0.066_{\pm 0.003}$ | $0.907_{\pm 0.004}$ | $-0.153_{\pm 0.009}$ |
| | 0.5 | $0.905_{\pm 0.009}$ | $-0.002_{\pm 0.004}$ | $0.922_{\pm 0.005}$ | $-0.018_{\pm 0.013}$ | $0.941_{\pm 0.003}$ | $-0.070_{\pm 0.003}$ | $0.937_{\pm 0.003}$ | $-0.162_{\pm 0.008}$ |
| | 0.2 | $0.911_{\pm 0.013}$ | $-0.002_{\pm 0.006}$ | $0.917_{\pm 0.008}$ | $-0.017_{\pm 0.011}$ | $0.947_{\pm 0.004}$ | $-0.074_{\pm 0.005}$ | $0.935_{\pm 0.005}$ | $-0.162_{\pm 0.011}$ |
| | 0.1 | $0.905_{\pm 0.010}$ | $-0.001_{\pm 0.005}$ | $0.917_{\pm 0.000}$ | $-0.025_{\pm 0.007}$ | $0.951_{\pm 0.005}$ | $-0.079_{\pm 0.004}$ | $0.951_{\pm 0.006}$ | $-0.177_{\pm 0.009}$ |
| | 0.01 | $0.911_{\pm 0.007}$ | $-0.003_{\pm 0.004}$ | $0.918_{\pm 0.005}$ | $-0.033_{\pm 0.017}$ | $0.958_{\pm 0.003}$ | $-0.087_{\pm 0.002}$ | $0.959_{\pm 0.003}$ | $-0.197_{\pm 0.003}$ |
| | 0.001 | $0.908_{\pm 0.176}$ | $-0.009_{\pm 0.042}$ | $0.921_{\pm 0.181}$ | $-0.044_{\pm 0.013}$ | $0.957_{\pm 0.008}$ | $-0.091_{\pm 0.003}$ | $0.959_{\pm 0.002}$ | $-0.221_{\pm 0.008}$ |
| FAIRREPROGRAM (HARD/PATCH) | 1.0 | $0.897_{\pm 0.012}$ | $-0.005_{\pm 0.004}$ | $0.905_{\pm 0.006}$ | $-0.009_{\pm 0.007}$ | $0.938_{\pm 0.003}$ | $-0.065_{\pm 0.014}$ | $0.931_{\pm 0.002}$ | $-0.154_{\pm 0.004}$ |
| | 0.5 | $0.905_{\pm 0.014}$ | $-0.006_{\pm 0.026}$ | $0.917_{\pm 0.006}$ | $-0.028_{\pm 0.007}$ | $0.932_{\pm 0.002}$ | $-0.062_{\pm 0.002}$ | $0.937_{\pm 0.004}$ | $-0.164_{\pm 0.006}$ |
| | 0.2 | $0.902_{\pm 0.013}$ | $-0.006_{\pm 0.017}$ | $0.909_{\pm 0.020}$ | $-0.025_{\pm 0.020}$ | $0.941_{\pm 0.003}$ | $-0.073_{\pm 0.004}$ | $0.945_{\pm 0.005}$ | $-0.166_{\pm 0.013}$ |
| | 0.1 | $0.900_{\pm 0.014}$ | $-0.005_{\pm 0.016}$ | $0.909_{\pm 0.008}$ | $-0.024_{\pm 0.011}$ | $0.948_{\pm 0.005}$ | $-0.079_{\pm 0.003}$ | $0.951_{\pm 0.002}$ | $-0.183_{\pm 0.008}$ |
| | 0.01 | $0.896_{\pm 0.013}$ | $-0.004_{\pm 0.010}$ | $0.918_{\pm 0.003}$ | $-0.035_{\pm 0.005}$ | $0.967_{\pm 0.007}$ | $-0.087_{\pm 0.004}$ | $0.955_{\pm 0.004}$ | $-0.192_{\pm 0.010}$ |
| | 0.001 | $0.907_{\pm 0.007}$ | $-0.008_{\pm 0.012}$ | $0.921_{\pm 0.000}$ | $-0.042_{\pm 0.001}$ | $0.955_{\pm 0.004}$ | $-0.089_{\pm 0.005}$ | $0.958_{\pm 0.005}$ | $-0.228_{\pm 0.013}$ |

Table 5: Numerical results with standard derivation on `Civil Comments` and `CelebA` with different trigger size, corresponding to Fig. 9. We evaluate the bias scores with different trigger word numbers (`Civil Comments`) and different trigger size (`CelebA`) with fixed adversary weight $\lambda$. All reported results are the average of three random runs. We report the negative DP and the negative EO scores correspondingly for the "Fairness" column.

| Method | Trigger Size | Civil Comments | | | | Trigger Size | CelebA | | | |
| | | Demographic parity | | Equalized odds | | | Demographic parity | | Equalized odds | |
| | | Accuracy | Fairness | Accuracy | Fairness | | Accuracy | Fairness | Accuracy | Fairness |
|---|---|---|---|---|---|---|---|---|---|---|
| ERM | - | $0.922_{\pm 0.004}$ | $-0.036_{\pm 0.025}$ | $0.923_{\pm 0.004}$ | $-0.054_{\pm 0.005}$ | - | $0.961_{\pm 0.004}$ | $-0.094_{\pm 0.002}$ | $0.961_{\pm 0.004}$ | $-0.231_{\pm 0.008}$ |
| FAIRREPROGRAM (SOFT/BORDER) | 20 | $0.890_{\pm 0.011}$ | $-0.001_{\pm 0.000}$ | $0.910_{\pm 0.005}$ | $-0.004_{\pm 0.001}$ | 30 | $0.914_{\pm 0.002}$ | $-0.054_{\pm 0.008}$ | $0.903_{\pm 0.005}$ | $-0.155_{\pm 0.005}$ |
| | 10 | $0.906_{\pm 0.004}$ | $-0.002_{\pm 0.001}$ | $0.906_{\pm 0.004}$ | $-0.009_{\pm 0.010}$ | 25 | $0.933_{\pm 0.003}$ | $-0.061_{\pm 0.006}$ | $0.917_{\pm 0.003}$ | $-0.162_{\pm 0.009}$ |
| | 5 | $0.917_{\pm 0.003}$ | $-0.002_{\pm 0.001}$ | $0.917_{\pm 0.004}$ | $-0.011_{\pm 0.010}$ | 20 | $0.939_{\pm 0.006}$ | $-0.070_{\pm 0.008}$ | $0.923_{\pm 0.007}$ | $-0.170_{\pm 0.008}$ |
| | 2 | $0.912_{\pm 0.002}$ | $-0.007_{\pm 0.001}$ | $0.921_{\pm 0.002}$ | $-0.038_{\pm 0.009}$ | 15 | $0.943_{\pm 0.004}$ | $-0.074_{\pm 0.06}$ | $0.951_{\pm 0.004}$ | $-0.189_{\pm 0.011}$ |
| | 1 | $0.917_{\pm 0.001}$ | $-0.011_{\pm 0.000}$ | $0.920_{\pm 0.000}$ | $-0.051_{\pm 0.002}$ | 10 | $0.951_{\pm 0.004}$ | $-0.081_{\pm 0.008}$ | $0.958_{\pm 0.008}$ | $-0.222_{\pm 0.012}$ |
| FAIRREPROGRAM (HARD/PATCH) | 20 | $0.890_{\pm 0.002}$ | $-0.001_{\pm 0.001}$ | $0.892_{\pm 0.001}$ | $-0.005_{\pm 0.005}$ | 70 | $0.912_{\pm 0.005}$ | $-0.048_{\pm 0.005}$ | $0.932_{\pm 0.004}$ | $-0.151_{\pm 0.006}$ |
| | 10 | $0.891_{\pm 0.009}$ | $-0.001_{\pm 0.003}$ | $0.901_{\pm 0.002}$ | $-0.016_{\pm 0.005}$ | 60 | $0.937_{\pm 0.008}$ | $-0.056_{\pm 0.004}$ | $0.947_{\pm 0.004}$ | $-0.160_{\pm 0.011}$ |
| | 5 | $0.897_{\pm 0.012}$ | $-0.005_{\pm 0.004}$ | $0.905_{\pm 0.006}$ | $-0.009_{\pm 0.007}$ | 50 | $0.935_{\pm 0.002}$ | $-0.061_{\pm 0.008}$ | $0.954_{\pm 0.004}$ | $-0.172_{\pm 0.012}$ |
| | 2 | $0.905_{\pm 0.007}$ | $-0.006_{\pm 0.005}$ | $0.911_{\pm 0.002}$ | $-0.030_{\pm 0.007}$ | 40 | $0.944_{\pm 0.006}$ | $-0.074_{\pm 0.010}$ | $0.959_{\pm 0.004}$ | $-0.191_{\pm 0.009}$ |
| | 1 | $0.913_{\pm 0.002}$ | $-0.012_{\pm 0.001}$ | $0.921_{\pm 0.003}$ | $-0.051_{\pm 0.001}$ | 30 | $0.958_{\pm 0.004}$ | $-0.091_{\pm 0.007}$ | $0.958_{\pm 0.002}$ | $-0.204_{\pm 0.013}$ |

`CelebA`) with the task of predicting the hair color, and evaluated on (ResNet-20, `CelebA`) with the task of predicting smiling. For both tasks, the attribute gender is chosen as the demographic information throughout the experiments. We can see that the trigger still has good transferability with different model architectures. Meanwhile, we find that the triggers are able to boost the fairness of the model in the task-transfer setting, but the accuracy is traded off more than the original setting.

## B.6 Experiments on Reprogramming Tabular Data

We show that FAIRREPROGRAM could also be applied to tabular data. For reprogramming, there are many ways to design triggers according to different tasks and requirements. Unlike NLP, where we append the trigger to the input or embeddings, the model for tabular data is sensitive to the input size. As the tabular data have a fixed input size, we can directly apply the additive trigger to the input data to keep the input dimension unchanged (i.e., adding a perturbation on the original input), just as we adopted in image domains in Fig. 1b. To verify our argument, we conducted additional experiments on the UCI Adult dataset [75] with a two-layer MLP model, and the results are shown in Fig. 13. Our method achieves comparable debiasing performance with the post-processing adversarial training method without modifying any model parameters. The results suggest that our method could effectively improve model fairness for tabular data.

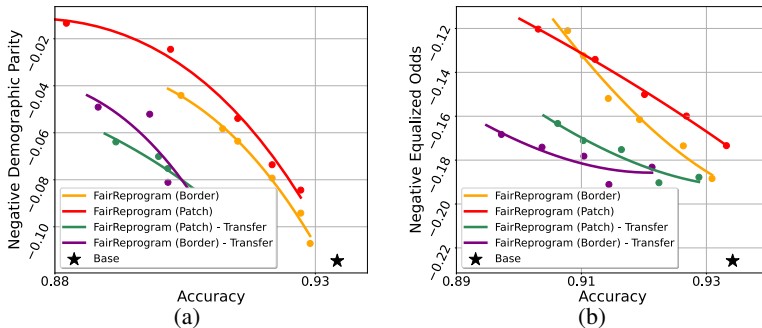

Figure 12: Results of the transferability experiment on `CelebA` dataset with different tasks and model architectures. In each figure, we compare the reprogramming in the transferred setting (curves denoted with 'transfer') with the reprogram directly trained on the target task. For transfer setting, the reprogram is optimized on the (ResNet-18, `CelebA`) with the task of predicting the hair color, and evaluated on (ResNet-20, `CelebA`) with the task of predicting smiling. For both tasks, the attribute gender is chosen as the demographic information throughout the experiments.

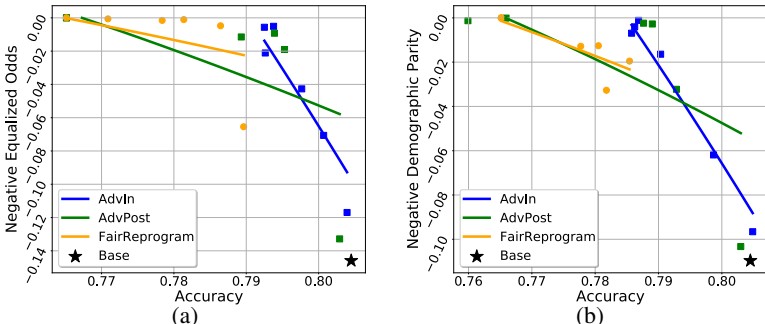

Figure 13: Results on `Adult`. We report the negative DP (left) and the negative EO (right) scores. For each method, we vary the trade-off parameter $\lambda$ (as shown in (3)) to record the performance. The closer a dot to the upper-right corner, the better the model is. We consider six different $\lambda$s for each method. The solid curve is the fitted polynomial with order 30.

## C  Theoretical Proofs

In this section, we will provide formal proofs to the claims and theorem in the main paper.

### C.1  Sufficient Statistics

We will show that $p_Y(\cdot|\boldsymbol{X}^{(y)})$ and $p_Z(\cdot|\boldsymbol{X}^{(z)})$ are the sufficient statistics of $\boldsymbol{X}^{(y)}$ and $\boldsymbol{X}^{(z)}$ respectively for inferring $Y$. Formally, what we need to show is

$$p(Y|\boldsymbol{X}^{(y)}) = p(Y|p_Y(\cdot|\boldsymbol{X}^{(y)})) \tag{9}$$

and

$$p(Y|\boldsymbol{X}^{(z)}) = p(Y|p_Z(\cdot|\boldsymbol{X}^{(z)})) \tag{10}$$

Eq. (9) is an identity. To show Eq. (10):

$$
\begin{aligned}
p(Y|\boldsymbol{X}^{(z)}) &= \mathbb{E}_{Z \sim p_Z(\cdot|\boldsymbol{X}^{(z)})}[p(Y|Z, \boldsymbol{X}^{(z)})] \\
&= \mathbb{E}_{Z \sim p_Z(\cdot|\boldsymbol{X}^{(z)})}[p(Y|Z, p_Z(\cdot|\boldsymbol{X}^{(z)}))] \\
&= \mathbb{E}_{Z \sim p_Z(\cdot|p_Z(\cdot|\boldsymbol{X}^{(z)}))}[p(Y|Z, p_Z(\cdot|\boldsymbol{X}^{(z)}))] \\
&= p(Y|p_Z(\cdot|\boldsymbol{X}^{(z)})).
\end{aligned}
$$

The second equality is because $Y$ and $\boldsymbol{X}^{(z)}$ are independent conditional on $Z$, so replacing $\boldsymbol{X}^{(z)}$ with any functions of $\boldsymbol{X}^{(z)}$ would not change the conditional probability. The third equality is implied from the identity $p(Z|\boldsymbol{X}^{(z)}) = p(Z|p_Z(\cdot|\boldsymbol{X}^{(z)}))$.

Using the sufficient statistics, it is very easy to show that $p(Y|\boldsymbol{X})$ is a special case of Eq. (6):

$$p(Y|\boldsymbol{X}) = p(Y|\boldsymbol{X}^{(y)}, \boldsymbol{X}^{(z)}) = p(Y|p_Y(\cdot|\boldsymbol{X}^{(y)}), p_Z(\cdot|\boldsymbol{X}^{(z)})).$$

## C.2 Proof to Thm. 1

We first provide the regularity conditions as stated in Thm. 1.

1. *Conditional Independence.* The features in $\boldsymbol{X}^{(y)}$ and $\boldsymbol{X}^{(z)}$ are independent and identically distributed conditional on $Y$ and $Z$ respectively.

$$p^{tr}(\boldsymbol{X}^{(y)}|Y) = \prod_t p^{tr}(\boldsymbol{X}_t^{(y)}|Y), \quad p^{tr}(\boldsymbol{X}^{(z)}|Z) = \prod_t p^{tr}(\boldsymbol{X}_t^{(z)}|Z). \tag{11}$$

2. *Infrequent Strong Demographic Features.* The probability of occurrence of features that are very strongly indicative against a certain demographic group is low. Formally $\forall z$, $\forall \varepsilon > 0$, $\exists \sigma > 0$, such that define

$$\mathcal{S}(\sigma) = \{\boldsymbol{x}^{(z)} \in \mathcal{X}^{(z)} : p(Z = z|\boldsymbol{X}^{(z)} = \boldsymbol{x}^{(z)}) \le \sigma\}, \tag{12}$$

we have

$$p(\boldsymbol{X}^{(z)} \in \mathcal{S}(\sigma)) \le \varepsilon. \tag{13}$$

3. *Continuous Classifier.* $h(\cdot, \cdot)$ is continuous with respect to both arguments.

With these assumptions, we will state the following lemma.

**Lemma 1.1.** *Consider the case where $Z$ takes on $K$ different values,* i.e., *there are $K$ demographic groups. Then*

$$\lim_{p^{tr}(Z=z|\boldsymbol{X}_0^{(z)}=\boldsymbol{\delta})\to 1} H(Z|h(p_Y^{tr}(\cdot|\boldsymbol{X}^{(y)}), p_Z^{tr}(\cdot|\boldsymbol{X}^{(z)} = \tilde{\boldsymbol{X}}^{(z)})), Y) = H(Z|h(p_Y^{tr}(\cdot|\boldsymbol{X}^{(y)}), c), Y), \tag{14}$$

*where $c$ is a $K$-dimensional one-hot vector with the $z$-th dimension equal to 1 and 0 elsewhere.*

*Proof.* According to Assumption 2 (Eq. (13)),

$$\forall \varepsilon > 0, \quad \exists 0 < \sigma < 1, \quad p(\boldsymbol{X}^{(z)} \in \mathcal{S}(\sigma)) \le \frac{\varepsilon}{4H(Z)}, \tag{15}$$

where $\mathcal{S}(\sigma)$ is defined in Eq. (12). On the other hand, consider the following composite function

$$H(Z|h(p_Y^{tr}(\cdot|\boldsymbol{X}^{(y)}), p_Z^{tr}(\cdot|\boldsymbol{X}^{(z)} = [\boldsymbol{\delta}, \boldsymbol{x}^{(z)}])), Y). \tag{16}$$

Note that this is *different* from $H(Z|h(p_Y^{tr}(\cdot|\boldsymbol{X}^{(y)}), p_Z^{tr}(\cdot|\boldsymbol{X}^{(z)} = \tilde{\boldsymbol{X}}^{(z)})), Y)$, which is essentially the expectation of Eq. (16) over different values of $\boldsymbol{x}^{(z)}$.

Since the conditional entropy is continuous and bounded, and $h(\cdot, \cdot)$ is continuous over both of its arguments with finite support, Eq. (16) is *uniformly* continuous with respect to $p_Z^{tr}(\cdot|\boldsymbol{X}^{(z)}))$. Therefore, given the aforementioned $\varepsilon$,

$$\exists 0 < \eta < 1, \quad \forall \boldsymbol{\delta}, \boldsymbol{x}^{(z)} \text{ s.t. } \|p_Z(\cdot|\boldsymbol{X}^{(z)} = [\boldsymbol{\delta}, \boldsymbol{x}^{(z)}]) - c\|_1 \le \eta$$
$$\Rightarrow \left| H(Z|h(p_Y^{tr}(\cdot|\boldsymbol{X}^{(y)}), p_Z^{tr}(\cdot|\boldsymbol{X}^{(z)} = [\boldsymbol{\delta}, \boldsymbol{x}^{(z)}])), Y) - H(Z|h(p_Y^{tr}(\cdot|\boldsymbol{X}^{(y)}), c), Y) \right| \le \frac{\varepsilon}{2}. \tag{17}$$

Now, divide the support of $\boldsymbol{X}^{(z)}$ into two disjoint sets. For notational conciseness, define

$$r(\boldsymbol{x}^{(z)}) = \max_{z' \ne z} \frac{p^{tr}(Z = z')p^{tr}(\boldsymbol{X}^{(z)} = \boldsymbol{x}^{(z)}|Z = z')}{p^{tr}(Z = z)p^{tr}(\boldsymbol{X}^{(z)} = \boldsymbol{x}^{(z)}|Z = z)}. \tag{18}$$

Then the two sets, denoted as $\mathcal{A}$ and $\mathcal{B}$ respectively, are divided according to whether $r(\boldsymbol{x}^{(z)})$ exceeds a threshold, *i.e.*

$$\mathcal{A} = \{\boldsymbol{x}^{(z)} \in \mathcal{X}^{(z)} : r(\boldsymbol{x}^{(z)}) \le \sigma^{-1} - 1\}, \quad \mathcal{B} = \{\boldsymbol{x}^{(z)} \in \mathcal{X}^{(z)} : r(\boldsymbol{x}^{(z)}) > \sigma^{-1} - 1\}. \tag{19}$$

$\forall \boldsymbol{x}^{(z)} \in \mathcal{A}$, define

$$\zeta = 1 - \left[ \frac{(1 - \eta/2)^{-1} - 1}{(K - 1)(\sigma^{-1} - 1)G} + 1 \right]^{-1}, \quad \text{where } G = \max_{z' \ne z} \frac{p(Z = z)}{p(Z = z')}. \tag{20}$$

Then we will show that

$$\forall \boldsymbol{x}^{(z)} \in \mathcal{A}, \forall \boldsymbol{\delta} \text{ s.t. } p^{tr}(Z = z|\boldsymbol{X}_0^{(z)} = \boldsymbol{\delta}) \ge 1 - \zeta \quad \Rightarrow \quad \|p_Z^{tr}(\cdot|\boldsymbol{X}^{(z)} = [\boldsymbol{\delta}, \boldsymbol{x}^{(z)}]) - c\|_1 \le \eta, \tag{21}$$

and hence Eq. (17) holds. This is because, according to the Bayesian rule,

$$p^{tr}(Z = z | \mathbf{X}_0^{(z)} = \boldsymbol{\delta}) = \frac{p^{tr}(Z = z) p^{tr}(\mathbf{X}_0^{(z)} = \boldsymbol{\delta} | Z = z)}{\sum_{z' \neq z} p^{tr}(Z = z') p^{tr}(\mathbf{X}_0^{(z)} = \boldsymbol{\delta} | Z = z')}. \tag{22}$$

Therefore

$$p^{tr}(Z = z | \mathbf{X}_0^{(z)} = \boldsymbol{\delta}) \geq 1 - \zeta \quad \Rightarrow \quad 1 + \sum_{z' \neq z} \frac{p^{tr}(Z = z') p^{tr}(\mathbf{X}_0^{(z)} = \boldsymbol{\delta} | Z = z')}{p^{tr}(Z = z) p^{tr}(\mathbf{X}_0^{(z)} = \boldsymbol{\delta} | Z = z)} \leq (1 - \zeta)^{-1}$$

$$\Rightarrow \quad \frac{p^{tr}(Z = z') p^{tr}(\mathbf{X}_0^{(z)} = \boldsymbol{\delta} | Z = z')}{p^{tr}(Z = z) p^{tr}(\mathbf{X}_0^{(z)} = \boldsymbol{\delta} | Z = z)} \leq (1 - \zeta)^{-1} - 1, \forall z' \neq z \tag{23}$$

$$\Rightarrow \quad \frac{p^{tr}(\mathbf{X}_0^{(z)} = \boldsymbol{\delta} | Z = z')}{p^{tr}(\mathbf{X}_0^{(z)} = \boldsymbol{\delta} | Z = z)} \leq G[(1 - \zeta)^{-1} - 1], \forall z' \neq z.$$

As a result,

$$p^{tr}(Z = z | \mathbf{X}^{(z)} = [\boldsymbol{\delta}, \mathbf{x}^{(z)}])^{-1} = 1 + \sum_{z' \neq z} \frac{p^{tr}(Z = z') p^{tr}(\mathbf{X}^{(z)} = \mathbf{x}^{(z)} | Z = z') p^{tr}(\mathbf{X}_0^{(z)} = \boldsymbol{\delta} | Z = z')}{p^{tr}(Z = z) p^{tr}(\mathbf{X}^{(z)} = \mathbf{x}^{(z)} | Z = z) p^{tr}(\mathbf{X}_0^{(z)} = \boldsymbol{\delta} | Z = z)}$$

$$\leq 1 + r(\mathbf{x}^z) \sum_{z' \neq z} \frac{p^{tr}(\mathbf{X}_0^{(z)} = \boldsymbol{\delta} | Z = z')}{p^{tr}(\mathbf{X}_0^{(z)} = \boldsymbol{\delta} | Z = z)} \tag{24}$$

$$\leq 1 + (\sigma^{-1} - 1) \sum_{z' \neq z} \frac{p^{tr}(\mathbf{X}_0^{(z)} = \boldsymbol{\delta} | Z = z')}{p^{tr}(\mathbf{X}_0^{(z)} = \boldsymbol{\delta} | Z = z)}$$

$$\leq 1 + (\sigma^{-1} - 1)(K - 1) G[(1 - \zeta)^{-1} - 1] = (1 - \eta/2)^{-1},$$

where the first line is implied from the Bayesian rule and assumption 1 (similar to Eq. (22)); the second line is implied from the definition of $r(\mathbf{x}^z)$ as in Eq. (18); the third line is due to the definition of set $\mathcal{A}$ as in Eq. (19) (note that the scope of Eq. (21) is confined to $\forall \mathbf{x}^{(z)} \in \mathcal{A}$); the last line is implied from Eq. (23) and the definition of $\zeta$ as in Eq. (20).

It then follows that

$$\|p_Z^{tr}(\cdot | \mathbf{X}^{(z)} = [\boldsymbol{\delta}, \mathbf{x}^{(z)}]) - c\|_1 = 1 - p^{tr}(Z = z | \mathbf{X}^{(z)}) + \sum_{z' \neq '} p^{tr}(Z = z' | \mathbf{X}^{(z)})$$

$$= 2(1 - p^{tr}(Z = z | \mathbf{X}^{(z)})) \tag{25}$$

$$\leq \eta,$$

where the first line is implied from the definition of the one-hot vector $c$ as well as the fact that the probability mass function is alwasy between 0 and 1; the second line is given by the fact that any probability mass functions sum to 1; and the last line is given by Eq. (24). This concludes the proof to Eq. (21).

Next, notice that

$$H(Z | h(p_Y^{tr}(\cdot | \mathbf{X}^{(y)}), p_Z^{tr}(\cdot | \mathbf{X}^{(z)} = \tilde{\mathbf{X}}^{(z)})), Y)$$

$$= \sum_{\mathbf{x}^{(z)} \in \mathcal{X}^{(z)}} H(Z | h(p_Y^{tr}(\cdot | \mathbf{X}^{(y)}), p_Z^{tr}(\cdot | \mathbf{X}^{(z)} = [\boldsymbol{\delta}, \mathbf{x}^{(z)}])), Y) p(\mathbf{X}^{(z)} = \mathbf{x}^{(z)})$$

$$= \sum_{\mathbf{x}^{(z)} \in \mathcal{A}} H(Z | h(p_Y^{tr}(\cdot | \mathbf{X}^{(y)}), p_Z^{tr}(\cdot | \mathbf{X}^{(z)} = [\boldsymbol{\delta}, \mathbf{x}^{(z)}])), Y) p(\mathbf{X}^{(z)} = \mathbf{x}^{(z)}) \tag{26}$$

$$+ \sum_{\mathbf{x}^{(z)} \in \mathcal{B}} H(Z | h(p_Y^{tr}(\cdot | \mathbf{X}^{(y)}), p_Z^{tr}(\cdot | \mathbf{X}^{(z)} = [\boldsymbol{\delta}, \mathbf{x}^{(z)}])), Y) p(\mathbf{X}^{(z)} = \mathbf{x}^{(z)}).$$

Thus

$$\left| H(Z | h(p_Y^{tr}(\cdot | \mathbf{X}^{(y)}), p_Z^{tr}(\cdot | \mathbf{X}^{(z)} - \tilde{\mathbf{X}}^{(z)})), Y) - H(Z | h(p_Y^{tr}(\cdot | \mathbf{X}^{(y)}), c), Y) \right|$$

$$= \sum_{\mathbf{x}^{(z)} \in \mathcal{A}} \left| H(Z | h(p_Y^{tr}(\cdot | \mathbf{X}^{(y)}), p_Z^{tr}(\cdot | \mathbf{X}^{(z)} = [\boldsymbol{\delta}, \mathbf{x}^{(z)}])), Y) - H(Z | h(p_Y^{tr}(\cdot | \mathbf{X}^{(y)}), c), Y) \right| p(\mathbf{X}^{(z)} = \mathbf{x}^{(z)})$$

$$+ \sum_{\mathbf{x}^{(z)} \in \mathcal{B}} \left| H(Z | h(p_Y^{tr}(\cdot | \mathbf{X}^{(y)}), p_Z^{tr}(\cdot | \mathbf{X}^{(z)} = [\boldsymbol{\delta}, \mathbf{x}^{(z)}])), Y) - H(Z | h(p_Y^{tr}(\cdot | \mathbf{X}^{(y)}), c), Y) \right| p(\mathbf{X}^{(z)} = \mathbf{x}^{(z)})$$

$$\tag{27}$$

In the following, we will bound the two terms respectively. For the first term in Eq. (27), Eq. (21) applies because $\boldsymbol{x}^{(z)} \in \mathcal{A}$. Therefore, according to Eq. (21) and (17), when $p^{tr}(Z = z | \boldsymbol{X}_0^{(z)} = \boldsymbol{\delta}) \geq 1 - \zeta$, we have

$$\sum_{\boldsymbol{x}^{(z)} \in \mathcal{A}} \left| H(Z | h(p_Y^{tr}(\cdot | \boldsymbol{X}^{(y)}), p_Z^{tr}(\cdot | \boldsymbol{X}^{(z)} = [\boldsymbol{\delta}, \boldsymbol{x}^{(z)}])), Y) - H(Z | h(p_Y^{tr}(\cdot | \boldsymbol{X}^{(y)}), c), Y) \right| p(\boldsymbol{X}^{(z)} = \boldsymbol{x}^{(z)})$$
$$\leq \sum_{\boldsymbol{x}^{(z)} \in \mathcal{A}} \frac{\varepsilon}{2} p(\boldsymbol{X}^{(z)} = \boldsymbol{x}^{(z)}) \leq \frac{\varepsilon}{2}. \tag{28}$$

For the second term, notice that when $\boldsymbol{x}^{(z)} \in \mathcal{B}$, $r(\boldsymbol{x}^{(z)}) > \sigma^{-1} - 1$ (according to Eq. (19)). So it follows that

$$p(Z = z | \boldsymbol{X}^{(z)} = \boldsymbol{x}^{(z)})^{(-1)} = 1 + \sum_{z' \neq z} \frac{p^{tr}(Z = z') p^{tr}(\boldsymbol{X}_0^{(z)} = \boldsymbol{\delta} | Z = z')}{p^{tr}(Z = z) p^{tr}(\boldsymbol{X}_0^{(z)} = \boldsymbol{\delta} | Z = z)} \tag{29}$$
$$\geq 1 + r(\boldsymbol{x}^{(z)}) \geq \sigma^{-1}$$

According to (15), this implies

$$p(\boldsymbol{X}^{(z)} \in \mathcal{B}) \leq p(\boldsymbol{X}^{(z)} \in \mathcal{S}(\sigma)) \leq \frac{\varepsilon}{4H(Z)}, \tag{30}$$

and further

$$\sum_{\boldsymbol{x}^{(z)} \in \mathcal{B}} \left| H(Z | h(p_Y^{tr}(\cdot | \boldsymbol{X}^{(y)}), p_Z^{tr}(\cdot | \boldsymbol{X}^{(z)} = [\boldsymbol{\delta}, \boldsymbol{x}^{(z)}])), Y) - H(Z | h(p_Y^{tr}(\cdot | \boldsymbol{X}^{(y)}), c), Y) \right| p(\boldsymbol{X}^{(z)} = \boldsymbol{x}^{(z)})$$
$$\leq \left[ H(Z | h(p_Y^{tr}(\cdot | \boldsymbol{X}^{(y)}), p_Z^{tr}(\cdot | \boldsymbol{X}^{(z)} = [\boldsymbol{\delta}, \boldsymbol{x}^{(z)}])), Y) + H(Z | h(p_Y^{tr}(\cdot | \boldsymbol{X}^{(y)}), c), Y) \right] p(\boldsymbol{X}^{(z)} = \boldsymbol{x}^{(z)})$$
$$\leq 2H(Z) p(\boldsymbol{X}^{(z)} = \boldsymbol{x}^{(z)}) \leq \frac{\varepsilon}{2}. \tag{31}$$

Plugging Eqs. (28) and (31) into Eq. (27), we can finally establish that $\forall \varepsilon > 0$, $\exists \zeta > 0$ (one possible $\zeta$ as defined in Eq. (20)), when $p^{tr}(Z = z | \boldsymbol{X}_0^{(z)} = \boldsymbol{\delta}) \geq 1 - \zeta$, we have

$$\left| H(Z | h(p_Y^{tr}(\cdot | \boldsymbol{X}^{(y)}), p_Z^{tr}(\cdot | \boldsymbol{X}^{(z)} - \tilde{\boldsymbol{X}}^{(z)})), Y) - H(Z | h(p_Y^{tr}(\cdot | \boldsymbol{X}^{(y)}), c), Y) \right| \leq \varepsilon. \tag{32}$$

Hence this concludes the proof to Lemma 1.1. $\qquad \square$

With Lemma 1.1, we are ready to prove Thm 1.

*Proof.* Note that

$$H(Z | h(p_Y^{tr}(\cdot | \boldsymbol{X}^{(y)}), c), Y) \geq H(Z | p_Y^{tr}(\cdot | \boldsymbol{X}^{(y)}), Y) = H(Z | Y). \tag{33}$$

The inequality sign is given by the data processing inequality; the equality is given by the fact that $Z$ and $\boldsymbol{X}^{(y)}$ are independent conditional on $Y$. On the other hand,

$$H(Z | h(p_Y^{tr}(\cdot | \boldsymbol{X}^{(y)}), c), Y) \leq H(Z | Y). \tag{34}$$

Combining Eqs. (33) and (34), we have

$$H(Z | h(p_Y^{tr}(\cdot | \boldsymbol{X}^{(y)}), c), Y) = H(Z | Y). \tag{35}$$

According to Lemma 1.1, when $p^{tr}(Z = z | \boldsymbol{X}_0^{(z)} = \boldsymbol{\delta}) \to 1$,

$$H(Z | h(p_Y^{tr}(\cdot | \boldsymbol{X}^{(y)}), p_Z^{tr}(\cdot | \boldsymbol{X}^{(z)} = \tilde{\boldsymbol{X}}^{(z)})), Y) \to H(Z | h(p_Y^{tr}(\cdot | \boldsymbol{X}^{(y)}), c), Y) = H(Z | Y). \tag{36}$$
$$\square$$

## C.3 Discussion on Feature Disentanglement Assumption

In Section 3.4, we made a simplifying assumption that all features could be divided into two disentangled groups, *i.e.*, $\boldsymbol{X} = [\boldsymbol{X}^y, \boldsymbol{X}^z]$, which are governed by the output label $Y$ and demographic information $Z$, respectively. The corresponding data generation process could be seen in Figure 2. On the other hand, however, if features are entangled in practice, we show that FAIRREPROGRAM can still provide false demographic information to overshadow the true demographics in Table 1.

## D  Broader Impact

Although there has been a proliferation of works in promoting ML fairness, most methods require training or finetuning the existing models to meet certain fairness notions. However, this could bring large computational and storage costs, low data efficiency, and model privacy issues with those large-scaled trained models.

Inspired by recent advances in model reprogramming techniques, we propose a new generic post-processing fairness learning framework. Specifically, we consider a fixed ML model and optimize a fairness trigger that is appended to the inputs with a min-max formulation. The proposed method enjoys a better fairness-accuracy trade-off compared with vast fairness promoting baselines with far less training costs.

Despite the effectiveness of our method, we note that our method still has some limitations. As a future research remark, our method still requires demographic annotations to remove biases, which could be hard to acquire in practice. It remains an open problem to develop a fairness-promoting technique without the use of demographics annotations.

We do not observe any potential negative societal impacts of our method. Instead, we believe that the outcome of our work could help enhance fairness of AI systems in a computationally-efficient and constraint-least manner. It can also provide broad positive impacts on diverse areas where AI techniques are applied.