# OpenReview forum: "Fairness Reprogramming"
_NeurIPS.cc/2022/Conference — NeurIPS 2022 Accept_

### Official Review · Reviewer_4SAR · 2022-06-28

**Rating:** 6
**Confidence:** 2
**Soundness:** 3 good
**Presentation:** 3 good
**Contribution:** 3 good

**Summary:**

Traditional approaches for fairness in neural networks suggest training or fine-tuning the entire weights of the network to achieve desired fairness criteria. This paper, instead, proposes a reprogramming based technique, called FairReprogram. Considering the neural network fixed, FairReprogram appends a global fairness trigger to the input to achieve fairness improvement with respect to existing fairness metrics. The results are backed up with theoretical analysis and experimental evaluations in NLP and CV datasets.

**Questions:**

Please address questions in the `weakness` of the paper above.

**Limitations:**

There is no negative societal impact of the work, as far as I know.

**Strengths And Weaknesses:**

Strength

- The paper is very well written. Statements are supported with examples. The paper is also well-motivated as reprogramming towards fairness improvement is less costly than retraining or fine-tuning with fairness objectives.

Weakness:

- How are correlations among features handled in FairReprogram?

- What is the intuition behind adding noise as fairness trigger, such as in patch trigger and border trigger? Does this mean demographic information is confined either in the border of the image or in a specific area of an image covered by the patch?

- In Figure 1, perhaps the captions are wrong as 1(a) should be about border trigger and vice-versa. A careful recheck is encouraged.

- Does the method extend to tabular data with fixed set of features in matrix form?

- A comparison with existing fairness improvement techniques such as pre-processing, in-processing, post-processing fairness algorithms should be discussed. In which family of fairness algorithm does this approach belong to?

---

> ### Author Response · Authors · 2022-08-02
> **Point-to-point Response to Reviewer 4SAR**
>
> Thank you very much for providing us with very constructive comments. In what follows, please see our responses.
>
> **Q1: How are correlations among features handled in FairReprogram?**
>
>
> **A1**: In the theoretical analysis, we made a simplifying assumption that the features are uncorrelated. However, this is just an assumption for the ease and brevity of our proof. In fact, if features do have correlations, our theoretical analysis will still hold – it can still be shown that the FairReprogram can provide false demographic info to overshadow the true one. The only difference from the case without correlations is that in the case with correlations among features, the trigger needs to provide even stronger false demographic cues to overshadow the additional demographic information reflected in the correlations among features. Moreover, our empirical results also verify that FairReprogram handles the correlations among features well, as can be shown by its superior performance on various datasets (Table 3), where correlations among features are ubiquitous. We will add this discussion to the paper.
>
>
> **Q2: What is the intuition behind adding noise as a fairness trigger, such as in patch trigger and border trigger? Does this mean demographic information is confined either in the border of the image or in a specific area of an image covered by the patch?**
>
> **A2**: When an image is appended with the fairness trigger, there will be two types of demographic cues. First, the original, true demographic cues that reside in the original image; second, the false demographic cues that reside in the trigger in the border/patch. The two cues can coexist and the false cues do not need to overlie the true cues. The key is that the false cues need to be strong enough so that the neural model, when presented with the two potentially conflicting cues, will go for the false one. This is entirely possible because the neural model has not seen the fairness trigger before so it cannot learn to ignore it. This intuition is also supported by our empirical analysis in Table 3, where the trigger is found to contain strong demographic cues. We will move Table 3 to the main paper and improve the clarity of the theoretical analysis sections.
>
> **Q3: In Figure 1, perhaps the captions are wrong as 1(a) should be about border trigger and vice-versa. A careful recheck is encouraged.**
>
> **A3**: Thank you very much for pointing out this typo and we will fix it in the revised version.
>
> **Q4: Does the method extend to tabular data with a fixed set of features in matrix form?**
>
> **A4**: Yes, fairness reprogramming can be applied to tabular data. There are many ways to design triggers. As the tabular data have a fixed input size, we can directly apply the **additive trigger** to the input data to keep the input dimension unchanged (i.e., adding a perturbation on the original input), just as we adopted in image domains (Figure 1). Thanks for pointing this out, we will include more discussion on trigger designs for different modalities of data in the revised version. To verify our argument, we applied our method to the tabular data and conducted additional experiments on the UCI Adult dataset with a two-layer MLP model, and the results are shown in this **[Figure](https://ibb.co/ssNyK7v)**. The results suggest that our method could effectively improve model fairness for tabular data. Our method achieves comparable debiasing performance with the post-processing adversarial training method without modifying any model parameters.
>
> **Q5: A comparison with existing fairness improvement techniques such as pre-processing, in-processing, post-processing fairness algorithms should be discussed. In which family of fairness algorithm does this approach belong to?**
>
> **A5**: Our work belongs to the post-processing category. The key difference between our method and pre/in-processing approaches lies in that our approach does not change the training data or interfere with the model training process.  In contrast, pre-processing methods need to alter the training data and therefore, need full access to the training data, model training process, and model parameters, which is a quite demanding requirement in real-world applications.  Our method focuses on the case, where we have no access to the training process at all but only the model.  Our method is also applicable to black-box settings (empirical results are shown in Appendix B), where we could correct a biased model without accessing the model parameters/gradients, which provides us a significant advantage over other in-processing approaches.  In addition, more empirical comparisons to other post-processing baselines can be found in Appendix B.

---

> ### Author Response · Authors · 2022-08-08
> **Genuine request for follow-up discussions**
>
> Dear Reviewer,
>
> We appreciate your efforts in reviewing our paper and your valuable comments. We have tried our best to address all your questions in detail. Could you please check our response, and let us know if you have further questions? Once again, thank you very much for your time, help and consideration.
>
> Best regards,
>
> Authors

---

> ### Author Response · Authors · 2022-08-09
> **Genuine request for follow-up discussions**
>
> Dear Reviewer 4SAR,
>
> We are very grateful for your valuable suggestions and insightful questions! We have tried our best to address your concerns. As there is only one day left for the author-reviewer discussions, we sincerely hope that you can provide us feedback before the discussion phase ends. We will be happy to know if there are any other concerns that we could address to better help you consider raising the score. Once again, thank you for your time and efforts in our work.
>
> Best regards,
>
> Authors

---

### Official Review · Reviewer_cTsj · 2022-07-11

**Rating:** 5
**Confidence:** 4
**Soundness:** 3 good
**Presentation:** 3 good
**Contribution:** 3 good

**Summary:**

This paper introduces a model reprogramming based fairness promoting method, consisting of a fixed ML model and optimizing a set of vectors concatenated on inputs to boost model fairness. An information-theoretic framework is also introduced to explain the rationales of fairness booster. Experiments on one NLP and another CV dataset demonstrate the utility of this approach.

**Questions:**

1. Many biased datasets are tabular represented, how does your work apply on tabular data? Appending additional dimension of vector directly?
2. Can you summarize why the appended information can always cut off the biased information path?
3. How do you distinguish this work with pre processing approaches?

**Limitations:**

See comments above.



**Strengths And Weaknesses:**

Strengths:
1. This paper tackles the fairness issue with reprogramming which is a relatively less explored area in fairness-aware learning.
2. The proposed method can finetune the pretrained model with computational efficiency.
3. Experiments show the effectiveness of proposed method.

Weaknesses:
1. The paper is not self-contained with several key information in Appendix for a good understanding of the claimed contributions. For example, without the understanding of "It can be shown (in Appendix C) that the posterior distributions.." and "it can be shown (in Appendix C) that the posterior distribution..", the claimed information-theoretic framework remains unclear.
2. With the above, it is not clear whether fairness can be really achieved by the proposed method.
3. Some related work on reprogramming fairness (for example "Reprogramming FairGANs with Variational Auto-Encoders: A New Transfer Learning Model") is not mentioned in the literature review. This work also related to incremental fairness but lacks of relevant discussion (for example "FAHT: An Adaptive Fairness-aware Decision Tree Classifier").
4. The size ratio for civil comments is 1/5 which does not align with the motivation that infeasible to retrain or finetune well-trained large-scale models.
5. The caption of Figure 1(a) and 1(b) are misplaced.

---

> ### Author Response · Authors · 2022-08-02
> **Point-to-point Response to Reviewer cTsj (Part 2/2)**
>
> **Q6: Can you summarize why the appended information can always cut off the biased information path?**
>
> **A6**: The trigger learned by the reprogram contains very strong demographic information and blocks the model from relying on the real demographic information from the input. This argument is both empirically verified by experiments (shown in Table 3) as well as theoretically proven in Sec. 3.4. Since the same trigger is attached to all the input, the uniformal demographic information contained in the trigger will weaken the dependence of the model on the true demographic information contained in the data, and thus improve the fairness of the pretrained model. Please kindly refer to our response to Q1 for a brief summary of how our algorithm works. We will move the relevant content to the main paper to improve the readability of the paper.
>
> **Q7: How do you distinguish this work with pre-processing approaches?**
>
> **A7**: Our work belongs to the post-processing category. The key difference between our method and the pre-processing approaches lies in that our approach does not need to change the training data or interfere with the model training process. In contrast, pre-processing methods need to alter the training data and therefore, need full access to the training data, model training process, and model parameters, which is a quite demanding requirement in real-world applications. Our method focuses on the case, where we have no access to the training process at all but only the model.

---

> ### Author Response · Authors · 2022-08-02
> **Point-to-point Response to Reviewer cTsj (Part 1/2)**
>
> We thank the reviewer for the constructive feedback. Please find our detailed responses below.
>
> **Q1: The paper is not self-contained with several key information in Appendix for a good understanding of the claimed contributions. With the above, it is not clear whether fairness can be really achieved by the proposed method.**
>
> **A1**: Thanks for your suggestion. Due to the page limit, we put the theoretical proof in the appendix and we will add more intuitive explanation as well as move the results shown in Table 3 (Appendix B) back to the main manuscript in the revised version to ensure better readability. As a brief summary of why our method works, essentially fairness triggers provide false and constant demographic info that tricks the biased model into believing all the input is from the same demographic group, which hinders the model from using the true demographic info to produce biased prediction.  As verified by the results in Table 3, the triggers all contain strong demographic cues. We will include all these discussions in the main paper. We also provide an intuitive interpretation of our theoretical proof below in A6.
>
> **Q2: Missing related works.**
>
> **A2**: Thanks very much for pointing out the relevant works. We will cite and discuss them in the related work in the revised version.
>
> **Q3: The size ratio for civil comments is 1/5 which does not align with the motivation that is infeasible to retrain or finetune well-trained large-scale models.**
>
> **A3**: First, we would like to bring to your attention that we have performed experiments with various data ratios as shown in Figure 4, where our method achieves significant improvement over baselines even in the extreme case with only 0.1% of the available data. This experiment can verify the ability of our algorithm to debias under extreme data scarcity. Moreover, we would like to clarify that the motivation of the proposed algorithm is to tackle the challenges in scenarios where access to the model parameters is restricted due to security, privacy or proprietary concerns (such as commercial API), rather than due to data limits. Therefore, we would need to test our algorithms under all different data ratios. That’s also the reason why we show the effectiveness of our proposed method in both the white-box and the black-box scenarios (Figure 10 in Appendix B), where even the model gradient is not accessible. The black-box setting is the most realistic application in the real-world scenario, while the white-box setting helps analyze and verify the efficacy of our approach. Given this motivation, we believe the experiments on civil comments constitute a relevant and valid test of our algorithm. We will modify our paper to clarify our motivation.
>
> **Q4: The captions of Figure 1(a) and 1(b) are misplaced.**
>
> **A4**: Thank you for pointing this out. We will fix this typo in the revised version.
>
> **Q5: Many biased datasets are tabularly represented. How does your work apply on tabular data? Appending additional dimension of vector directly?**
>
> **A5**: Fairness reprogramming can be applied to tabular data. For reprogramming, there are many ways to design triggers according to different tasks and requirements. Unlike NLP, where we append the trigger to the input or embeddings, the model for tabular data is sensitive to input size. As the tabular data have a fixed input size, we can directly apply the **additive trigger** to the input data to keep the input dimension unchanged (i.e., adding a perturbation on the original input), just as we adopted in image domains (Figure 1). Thanks for pointing out this possible application scenario for our method. To verify our argument, we applied our method to the tabular data and conducted additional experiments on the UCI Adult dataset with a two-layer MLP model, and the results are shown in this **[Figure](https://ibb.co/ssNyK7v)**. The results suggest that our method could effectively improve model fairness for tabular data. Our method achieves comparable debiasing performance with the post-processing adversarial training method without modifying any model parameters.

---

> ### Author Response · Authors · 2022-08-08
> **Genuine request for follow-up discussions**
>
> Dear Reviewer,
>
> We appreciate your efforts in reviewing our paper and your valuable comments. We have tried our best to address all your questions in detail. Could you please check our response, and let us know if you have further questions? Once again, thank you very much for your time, help and consideration.
>
> Best regards,
>
> Authors

---

> ### Author Response · Authors · 2022-08-09
> **Genuine request for follow-up discussions**
>
> Dear Reviewer cTsj,
>
> We are very grateful for your valuable suggestions and insightful questions! We have tried our best to address your concerns. As there is only one day left for the author-reviewer discussions, we sincerely hope that you can provide us feedback before the discussion phase ends. We will be happy to know if there are any other concerns that we could address to better help you consider raising the score. Once again, thank you for your time and efforts in our work.
>
> Best regards,
>
> Authors

---

### Official Review · Reviewer_4uyL · 2022-07-11

**Rating:** 5
**Confidence:** 4
**Soundness:** 3 good
**Presentation:** 3 good
**Contribution:** 3 good

**Summary:**

This paper discusses how to learn a constant feature that can be concatenated to a frozen pre-trained embedding to promote statistical group fairness in a downstream classifier. These extra features, called the “fairness trigger”, are jointly trained with the prediction model with the aim of inducing a demographic parity or equalized odds property in the final solution.

**Questions:**

* I am slightly concerned that the in/post-processing methods could be doing worse simply because of the instability of adversarial training, or due to overfitting given the limited number of data available during fine tuning for the post-processing method [line 244]. I would suggest adding an MMD baseline as an additional baseline (e.g. as applied in https://arxiv.org/abs/1511.00830). This would hopefully be more stable, but may also suffer from overfitting.
* In my opinion, the claim of achieving “lower bias scores over two fairness criteria in the CelebA dataset” [lines 54–55] is not meaningful without defining the prediction task, since many different tasks (related to fairness and otherwise) have been proposed that use the crowdsourced attributes of CelebA. The authors do a good job of providing this context in Section 4 [lines 219–200]; I would suggest adding a basic description of the task to Section 1 wherever the results are summarized.
* I’m unclear on why the JTT paper [12] is cited as a paper that “[learns] task-specific embedding prompts concatenated to the inputs” [lines 36–37]. My understanding of this paper is that it uses the error cases of an ERM reference model to train a robust model via importance weighting. Could the authors clarify the citation (in this context) or remove it?
* I was interested in the experiment showing that the method is not overly sensitive to random seed [lines 291–300]. Calling this a “transfer” is a bit generous—there is no notion of a distribution shift or new prediction task, or a change in the model architecture—but it is good to know that a learned prompt can be reused if the model is retrained on the same data. The claim at the end of the paper that the method “enjoys great transferability” [line 356] seems like an overstatement. I would be especially concerned that adding a single constant feature could suffer from covariate/distribution shift, which was not examined.
* In Figure 1 it seems that the captions and images should be swapped. I.e. Fig 1a is the border trigger, not the patch trigger, if I understand correctly.


**Limitations:**

Yes, the authors discuss several limitations in the appendix. I would encourage including them in the main paper.


**Strengths And Weaknesses:**

Strengths
* Demonstrates that reprogramming techniques can be adapted to target group fairness metrics in CV and NLP contexts.
Interesting qualitative analysis of the resulting models [Figs 6, 7] suggesting that the learned concatenated constant features cause the downstream classifier to focus less on spurious features (in this case, features that indicate demographic membership).
* Provides a conceptual proof of concept for reprogramming in a simple bag-of-words classification setting where data are generated according to an anti-causal graphical model [Figure 2].
Weaknesses
* I found the proposed method interesting and I think the neurips community will as well. My main complaint is that the paper, in my opinion, oversells this approach as strictly superior to previous methods. When we apply the fairness trigger in the original input space (e.g. CV), don’t we still need to backpropagate through the entire network to learn the fairness trigger? If yes, there could be a memory savings as we don’t tune as many parameters, but it’s not clear to me that the method would train substantially faster than a finetuning baseline. On the other hand I can see why you would save on compute when applying the fairness trigger *in* the embedding space (e.g. NLP)
* The paper mainly compares against fine tuning methods, but there are other feasible approaches such as adding a linear probe on the frozen embedding (which I don’t believe is considered the experiments). When we teach linear regression we often bring up two alternate parameterizations: (a) \hat y = x^T W + b and (b) \hat y = (\tilde x)^T W. These are equivalent if we define \tilde x as x with an extra constant feature. By analogy, I wonder if there is some equivalence between the fairness trigger (learned bias) and a linear probe/transformation applied on the embedding (learned projection). If the fairness trigger is learning a constant perturbation along some subspace with lots of demographic information, is it possible that a similar solution could be found by simply projecting away that subspace using a linear probe? It seems possible to me, and makes me wonder if the experimental gains are due to the fact that the fairness trigger optimizes a lower dimensional parameter space that would be less prone to overfitting given the limited data available for post-processing. Even if this were the case (might be possible to find out by adding linear probes as baselines in the experiments) the paper would be interesting, but might suggest that fairness reprogramming is best under a constrained data budget, not best in general.

---

> ### Author Response · Authors · 2022-08-02
> **Point-to-point response to reviewer 4uyl**
>
> We greatly appreciate your thoughtful comments! Please see our response to your questions and concerns as below.
>
> **Q1: Don’t we still need to backpropagate through the entire network to learn the fairness trigger? If yes, there could be a memory savings as we don’t tune as many parameters, but it’s not clear to me that the method would train substantially faster than a fine-tuning baseline.**
>
> **A1**: We would like to clarify that the main motivation for the fairness reprogramming algorithm is not to improve computational efficiency, but to resolve the challenges in many real-world applications where access to the model parameters is restricted, and therefore it is impossible to directly modify the model towards the fairness goals. That being said, we totally agree that the proposed method would not train substantially faster than the fine-tuning baseline and we do not intend to claim it does. It may still train slightly faster because of the reduced tuning parameters but that is a bit outside the scope and our claimed contributions of this paper. We will modify our paper to make this clearer.
>
> **Q2: If the fairness trigger is learning a constant perturbation along some subspace with lots of demographic information, is it possible that a similar solution could be found by simply projecting away that subspace using a linear probe?**
>
> **A2**: Firstly, we agree that for certain simple models, the reprogramming method is equivalent to adding a linear probe. Specifically, if the model is a simple MLP, a trigger added to the input can be easily regarded as appending a bias term to the first layer. Nevertheless, similar conclusions can not be extended to transformers or convolutional layers as used in the NLP and CV domain in our paper, since their functions are more complex and cannot be represented by simple linear transformations. The reprogramming method still has a stronger representation power in this case. Moreover, please kindly be reminded that the motivation of fairness reprogramming is to resolve fairness tuning without having access to the model parameters. Under this scenario, linear probe insertion is less applicable, whereas our method remains a feasible solution with decent representation power. Nevertheless, we greatly appreciate this inspiring question and we will regard it as an interesting topic for future research.
>
> **Q3: I am slightly concerned that the in/post-processing methods could be doing worse simply because of the instability of adversarial training, or due to overfitting given the limited number of data available during fine tuning for the post-processing method. I would suggest adding an MMD baseline as an additional baseline.**
>
>
> **A3**: Thank you for your suggestion! Following your suggestion, we conducted some additional experiments and introduced the suggested MMD baseline as used in https://arxiv.org/abs/1511.00830. Please refer to the results shown in this **[Figure](https://ibb.co/CsKQJ5B)**. As we can see, our proposed method still outperforms the MMD baselines, which can alleviate the concern that fairness reprogramming has a better performance simply because of the instability of adversarial training of the baselines. We will add the new baseline to our pool of standard baselines in the paper.
>
> **Q4: I would suggest adding a basic description of the task to Section 1 wherever the results are summarized.**
>
>
> **A4**: Thanks for your suggestion and we will add the task description in the introduction section as well in the revised version.
>
>
> **Q5: Could the authors clarify the citation (in this context) or remove it?**
>
> **A5**: We thank you for pointing out the typo here, and we would modify it in the revision.
>
> **Q6: The claim at the end of the paper that the method “enjoys great transferability” [line 356] seems like an overstatement. I would be especially concerned that adding a single constant feature could suffer from covariate/distribution shift, which was not examined.**
>
> **A6**: Thanks for pointing this out. In our previous transfer experiments, we adopted different models with the same architecture. As the reviewer suggested, we conduct additional experiments where we transfer the trigger from ResNet18 to a different architecture (ResNet20s). We also change the training task from predicting the hair color to predicting smiling/not smiling. The results are shown in this **[Figure](https://ibb.co/QPSdfB3)**. We can see that the trigger still has good transferability with different model architectures. Meanwhile, we find that the triggers are able to boost the fairness of the model in the task-transfer setting, but the accuracy is traded off more than the original setting. We will add the new experiments to the paper, as well as modify the paper with a more precise claim and a more detailed discussion.
>
> **Q7: In Figure 1 it seems that the captions and images should be swapped.**
>
> **A7**: Thanks a lot for pointing this out and we will fix it in the revised version.

---

> > ### Comment · Reviewer_4uyL · 2022-08-05
> > **a few small questions**
> >
> > Thank you for the considered response.
> >
> > I think I better understand the motivation now. The response indicates that the authors are not claiming a substantial speedup w.r.t. finetuning, but the current abstract still states that the proposed method enjoys "far less training cost" than retraining-based methods. I would suggest the language in the abstract be changed to make any claims about the proposed method as specific as possible. Correct me if I misunderstand finetuning as a type of retraining method (not retraining from scratch, but none-the-less updating model parameters).
> >
> > I'm not sure I agree about linear probes (applied on top of a fixed embedding) assume more access than reprogramming in the embedding space (e.g. in the NLP domain). It's possible I'm missing something basic, but could you expand on this point?
> >
> > Thank you for providing the new baseline. Are you sure these results are for Celeb-A? The shape of the pareto curves and legend seem to indicate the results come from civil comments.
> >
> > It would be helpful to either (a) see all baselines with the proposed method on the same plot or (b) fix the axes of the new MMD plot to match those of the original plot.

---

> > > ### Author Response · Authors · 2022-08-06
> > > **Response to the follow-up questions.**
> > >
> > > Thank you very much for your response and the follow-up questions. Please see our point-to-point answer to your questions below.
> > >
> > > **Q1: I would suggest the language in the abstract be changed to make any claims about the proposed method as specific as possible. Correct me if I misunderstand finetuning as a type of retraining method (not retraining from scratch, but none-the-less updating model parameters).**
> > >
> > > A1: Thanks very much for your kind suggestions. Yes, you are correct about the interpretation of finetuning used in our paper and we will revise the description in the revised version to make our motivation clearer.
> > >
> > > **Q2: I'm not sure I agree about linear probes (applied on top of a fixed embedding) assume more access than reprogramming in the embedding space (e.g. in the NLP domain). It's possible I'm missing something basic, but could you expand on this point?**
> > >
> > > A2: We are sorry that we may have a misunderstanding on your question. We are not sure whether “the fixed embedding which the linear probes are applied on top of” refers to the input embeddings or the last model hidden layer output. For both cases, however, the accessibility of the embeddings are always necessary when **applying the linear probes**, which could be infeasible in practice. For example, let’s consider a black-box NLP model whose parameters and architecture are transparent to users and only the output can be provided for a given input. The linear projection could not be applied due to the lack of access to embeddings. By contrast, fairness reprogramming still works as it only appends the trigger into the input sentences to re-purpose the model.
> > >
> > > Besides, back to the original question, we agree that our reprogramming method is equivalent to adding a linear transformation directly to the inputs in some simple cases like tabular data. We conducted additional experiments on the UCI Adult dataset with a two-layer MLP. An additive trigger is added to the original inputs with the input dimension unchanged, *i.e.*, $\tilde{x}=m \circ x+\delta$, where $m$ is a multi-dimensional binary mask and $\delta$ is the trigger. The **[results](https://ibb.co/ssNyK7v)** show that our method is comparable with the post-processing adversarial training baseline, which empirically demonstrates the equivalence. We believe such a discussion may provide a very valuable insight on how our method works beyond our conceptual proof in Section 3.4.
> > >
> > > We will add all the discussions above to our revised version and we truly appreciate your illuminating question!
> > >
> > > **Q3: Thank you for providing the new baseline. Are you sure these results are for Celeb-A? The shape of the pareto curves and legend seem to indicate the results come from civil comments. It would be helpful to either (a) see all baselines with the proposed method on the same plot or (b) fix the axes of the new MMD plot to match those of the original plot.**
> > >
> > > A3: Thanks for pointing this out! We are sorry there was a typo on the dataset name and we have corrected it to Civil Comment. According to your suggestion by putting all the baseline methods on the same plot, we have updated the results in this **[Figure](https://ibb.co/3vXzcn5)**. To improve the readability, we also provide another version in this **[Figure](https://ibb.co/Xyd4GWT)**, where all the data points are removed and only the curves remain. We hope the new plots can alleviate your concern that fairness reprogramming has a better performance simply because of the instability of adversarial training of the baselines. We will also update this figure in the revision.
> > >
> > > We hope our responses could address your questions. If you have additional comments, please feel free to let us know. We will try our best to resolve them.

---

> > > > ### Comment · Reviewer_4uyL · 2022-08-09
> > > > **about NLP setup**
> > > >
> > > > For the NLP setting, you say that linear projections would require access to text embeddings, whereas "fairness reprogramming still works as it only appends the trigger into the input sentences". When I look at the paper, in lines 258--260 I see that the perturbation is defined as a $\delta_i = E v_i$ where $E$ is the BERT embedding. From this I conclude that perturbations are optimized in the embedding space and require the embedding function. Am I missing something...?

---

> > > > > ### Author Response · Authors · 2022-08-09
> > > > > **Thanks for your prompt response and further clarification**
> > > > >
> > > > > Thank you for your rapid responses! We are sorry for the possible confusion and we make the further clarification below.
> > > > >
> > > > > Our method consists of two phases, namely to **obtain** the trigger first and then **apply** it. We agree that we need to use the embeddings for **obtaining** the trigger $\delta$ with $\delta_i=Ev_i$ in our current training method. On the other hand, our method does not require access to embeddings for **applying** the reprogramming compared to linear projections, supposing the trigger has already been obtained. As indicated by lines 264-267, the token selection vector $v_i$ is one-hot in *FairReprogram (HARD)*, so each $v_i$ corresponds to a specific word in the vocabulary. That being said, for applying the *FairReprogram (HARD)* during the inference phase, we could simply add the text suffix indicated by $v$ into the inputs, which is equivalent to appending $\delta=Ev$ in the embedding space. It is also worth mentioning that *FairReprogram (HARD)* has been shown to enjoy great transferability in Figure 5 of the paper, so one potential application is to obtain the triggers with a substitute model where embeddings are known and apply it to the target model of interest. We apologize for the confusion and we would make it clearer in our revised version!
> > > > >
> > > > > Besides, we would like to mention that it is also possible to optimize the fairness triggers with query-based methods so that the embeddings are not necessary for **obtaining** triggers either. Such methods have shown great success in generating adversarial attacks for black-box NLP models [1, 2, 3], which is similar to our setting. We thank you for your insightful question and we leave this for our future research.
> > > > >
> > > > > Again, thank you for your response and if you have any questions, we are more than happy to address them.
> > > > >
> > > > > > [1] Li, Linyang et al. “BERT-ATTACK: Adversarial Attack against BERT Using BERT.” ArXiv abs/2004.09984 (2020): n. Pag.
> > > > > >
> > > > > > [2]Jin, Di et al. “Is BERT Really Robust? A Strong Baseline for Natural Language Attack on Text Classification and Entailment.” AAAI (2020).
> > > > > >
> > > > > > [3] Garg, Siddhant and Goutham Ramakrishnan. “BAE: BERT-based Adversarial Examples for Text Classification.” ArXiv abs/2004.01970 (2020): n. pag.

---

> > > > > > ### Comment · Reviewer_4uyL · 2022-08-09
> > > > > > **obtaining vs applying**
> > > > > >
> > > > > > I see what you mean. In my mind, the obtaining/applying distinction isn't especially meaningful in terms of what level of access is assumed for the proposed method. For example you said "consider a black-box NLP model" not "consider a black-box NLP model where the fairness trigger has previously been obtained". This trigger needs to be obtained somehow, and it seems that we all agree that the embedding function is required for this, so I would say that fairness reprogramming (as proposed) is not a black-box method.
> > > > > >
> > > > > > I agree that the idea of training the fairness trigger using a separate architecture/dataset, then transferring it to a target model, would be a more compelling story. Recall that in my original review I expressed concern that describing the investigating robustness-to-random-seed as "enjoying great transferability" may be an overstatement. However, what you are not describing in the responses sounds more interesting in terms of a transfer problem and could be set up in a more authentically black-box way. I would also agree that the connection to adversarial attacks becomes more important if you were to take this angle. Data poisoning, including transferability thereof [Zhu et al 2019], is another related field to look into.
> > > > > >
> > > > > > I appreciate the authors staying engaged throughout the discussion period. I see the author-reviewer discussion as an opportunity to clarify the reviewer's understanding of what the paper claims and how those claims are substantiated. Answering the reviewer questions can certainly help clarify any misunderstandings, but these clarifications don't necessarily merit an increase to the score, which is ultimately reflects my opinion of the paper's quality, potential impact, and relevance to the conference. In my original review I expressed a concern that this paper oversells the approach. Even after the author-reviewer discussion, this is still a concern for me.
> > > > > >
> > > > > > References:
> > > > > > [Zhu et al 2019] https://arxiv.org/abs/1905.05897

---

> > > ### Author Response · Authors · 2022-08-08
> > > **Genuine request for follow-up discussions**
> > >
> > > Dear Reviewer,
> > >
> > > We appreciate your efforts in reviewing our paper and your valuable comments. We have tried our best to address all your questions in detail. Could you please check our response, and let us know if you have further questions? Once again, thank you very much for your time, help and consideration.
> > >
> > > Best regards,
> > >
> > > Authors

---

> > > ### Author Response · Authors · 2022-08-09
> > > **Genuine request for follow-up discussions**
> > >
> > > Dear Reviewer 4uyl,
> > >
> > > We are very grateful for your valuable suggestions and insightful questions! We have tried our best to address your concerns. As there is only one day left for the author-reviewer discussions, we sincerely hope that you can provide us feedback before the discussion phase ends. We will be happy to know if there are any other concerns that we could address to better help you consider raising the score. Once again, thank you for your time and efforts in our work.
> > >
> > > Best regards,
> > >
> > > Authors

---

### Meta-Review · Area_Chair_yGBU · 2022-08-23

**Recommendation:** Accept
**Confidence:** Less certain

**Metareview:**

Overall the reviews are more or less positive towards weak accept. While there are some remaining concerns (e.g., a wording in the abstract), I think many of the raised concerns are addressed properly and some of them are checked. Hence, I believe this paper is worth being published.

**Award:**

No

---

### Decision · Program_Chairs · 2022-09-14

Accept